# Aerosol hygroscopic growth, contributing factors, and impact on haze events in a severely polluted region in northern China

Jun Chen[1], Zhanqing Li[1,2], Min Lv[3], Yuying Wang[1], Wei Wang[1], Yingjie Zhang[4],

Haofei Wang[5,6], Xing Yan[1], Yele Sun[4], Maureen Cribb[2]

[1]State Key Laboratory of Remote Sensing Science, College of Global Change and Earth System Science, Beijing Normal University, Beijing 100875, China
[2]Department of Atmospheric and Oceanic Sciences and ESSIC, University of Maryland, College Park, Maryland, USA
[3]School of Geographic Science, Nantong University, Nantong 226000, China
[4]State Key Laboratory of Atmospheric Boundary Layer Physics and Atmospheric Chemistry, Institute of Atmospheric Physics, Chinese Academy of Sciences, Beijing 100029, China
[5]College of Resource Environment and Tourism, Capital Normal University, Beijing, 100048, China
[6]State Environment Protection Key Laboratory of Satellite Remote Sensing, Institute of Remote Sensing and Digital Earth, Chinese Academy of Sciences, Beijing, 100101, China

*Correspondence to*: Zhanqing Li (zli@atmos.umd.edu) and Yuying Wang (wang.yuying@mail.bnu.edu.cn)

**Abstract:**

This study investigates the impact of the aerosol hygroscopic growth effect on haze

events in Xingtai, a heavily polluted city in the central part of the North China Plain,

using a large array of instruments measuring aerosol optical, physical, and chemical

properties. Key instruments used and measurements made include the Raman lidar for

atmospheric water vapor content and aerosol optical profiles, the PC-3016A GrayWolf

six-channel handheld particle/mass meter for atmospheric total particulate matter (PM)

that have diameters less than 1 μm and 2.5 μm ($PM_1$ and $PM_{2.5}$, respectively), the

aerosol chemical speciation monitor (ACSM) for chemical components in $PM_1$, and the

hygroscopic tandem differential mobility analyzer (H-TDMA) for aerosol

hygroscopicity. The changes in $PM_1$ and $PM_{2.5}$ agreed well with that of the water vapor

content due to the aerosol hygroscopic growth effect. Two cases were selected to

further analyze the effects of aerosol hygroscopic growth on haze events. The lidar-

estimated hygroscopic enhancement factor for the aerosol backscattering coefficient

during a relatively clean period (Case I) was lower than that during a pollution event

(Case II) with similar relative humidity (RH) levels of 80–91%. The Kasten model was

used to fit the aerosol particle hygroscopic growth factor whose parameter $b$ differed

considerably between the two cases, i.e., 0.1000 (Case I) versus 0.9346 (Case II). The

aerosol acidity value calculated from ACSM data for Case I (1.35) was less than that

for Case II (1.50) due to different amounts of inorganics such as $NH_4NO_3$, $NH_4HSO_4$,

and $(NH_4)_2SO_4$. Model results based on H-TDMA data showed that aerosol hygroscopic

growth factors in each size category (40, 80, 110, 150, and 200 nm) at different RH

levels (80–91%) for Case I were lower than those for Case II. For similar ambient RH levels, the high content of nitrate facilitates the hygroscopic growth of aerosols, which may be a major factor contributing to heavy haze episodes in Xingtai.

**Key words:** Raman lidar; aerosol hygroscopic growth; water content; haze; remote sensing

## 1. Introduction

Aerosols, as solid or liquid particles suspended in the air, help regulate Earth's climate mainly by directly scattering or absorbing incoming radiation, or indirectly changing cloud optical and microphysical properties (IPCC, 2013). Many studies suggest that aerosols have a direct impact on human health (Araujo et al., 2008; Anenberg et al., 2010; Liao et al., 2015; Li et al., 2017). For example, exposure to fine airborne particulates is linked to increased respiratory and cardiovascular diseases (Hu et al., 2015). Atmospheric aerosols can also reduce visibility. Poor visibility is not only detrimental to human health but also hazardous to all means of transportation (Zhang et al., 2010; Zhang et al., 2018).

Poor visibility is caused by the presence of atmospheric aerosols whose loading depends on both emission and meteorology. The increase in anthropogenic emissions directly affects the formation of haze, such as biomass burning, and factory and vehicle emissions (Watson, 2002; Sun et al., 2006; Q. Liu et al., 2016; Qu et al., 2018). During some major events like the 2008 Summer Olympic Games, drastic measures were taken to reduce emissions which led to a significant improvement in air quality (Huang et al., 2014; Shi et al., 2016; Y.-Y. Wang et al., 2017). This attests to the major role of emissions in air quality. Surface solar radiation and weather such as wind conditions also affect aerosol pollution (Yang et al., 2015). It has been widely known that aerosols interact with the planetary boundary layer (PBL; Quan et al., 2013; Li et al., 2017; Qu et al., 2018; Su et al., 2018). More aerosols reduce surface solar radiation, resulting in a more stable PBL which enhances the accumulation of pollutants within the PBL. Numerous studies have highlighted that the diurnal evolution of the PBL is crucial to the formation of air pollution episodes (Tie et al., 2015; Amil et al., 2016; Kusumaningtyas and

1 Aldrian, 2016; Li et al., 2017; Qu et al., 2018). Besides feedbacks, the stability of the PBL

affects the dispersion of pollutants.

Aerosol hygroscopicity also significantly affects visibility due to the swelling of aerosols

(Jeong et al., 2007; Wang et al., 2014). A number of studies have shown that aerosol

hygroscopic growth can accelerate the formation and evolution of haze pollution in the North

China Plain (NCP; e.g., Quan et al., 2011; Liu et al., 2013; Wang et al., 2014; Yang et al., 2015).

There are many ways to measure aerosol hygroscopicity. A widely used parameter, the aerosol

particle size hygroscopic growth factor (GF), is defined as the ratio of the wet particle diameter

($D_{p,wet}$) at a high relative humidity (RH) to the corresponding dry diameter ($D_{p,dry}$). The GF at

a certain particle size can be detected by a hygroscopicity tandem differential mobility analyzer

(H-TDMA; e.g., Liu et al., 1978; Swietlicki et al., 2008; Y.-Y. Wang et al., 2017). In general,

the H-TDMA system mainly consists of two differential mobility analyzer (DMA) systems and

one condensation particle counter (CPC). The DMA is first used to select particles at a specific

size, and the second DMA and the CPC are used to measure the size distribution of humidified

particles. Another instrument known as the differential aerosol sizing and hygroscopicity

spectrometer probe (DASH-SP) can also measure the GF at different RHs (Sorooshian et al.,

2008). The DASH-SP couples one DMA and an optical particle size spectrometer (OPSS). The

dry size-dependent particles are selected by the DMA, then exposed to different RH

environments and finally measured in the OPSS (Sorooshian et al., 2008; Rosati et al., 2015).

The aerosol optical hygroscopic enhancement factor [$f$(RH)] has also been employed to

investigate aerosol hygroscopicity, which is defined as the ratio of aerosol optical properties

(aerosol extinction, scattering or backscattering coefficients) between wet and dry conditions

(Kotchenruther et al., 1998). Two tandem nephelometers are used to measure *f*(RH) (e.g., Covert et al., 1972; Feingold and Morley, 2003; Titos et al., 2018). One nephelometer measures the aerosol optical properties of dry ambient aerosols at RH < 40%, and another measures that of wet aerosols at different RHs adjusted by a humidifier placed between them (Koloutsou-Vakakis et al., 2001; Titos et al., 2018). MacKinnon (1969) was the first to find that the lidar backscattering signal is affected by environmental RHs. Later studies have demonstrated the possibility of using the lidar to observe aerosol hygroscopic growth (Tardif et al., 2002; Pahlow et al., 2006; Veselovskii et al., 2009; Di Girolamo et al., 2012; Fernández et al., 2015; Granados-Muñoz et al., 2015; Lv et al., 2017; Bedoya-Velásquez et al., 2018). Compared with tandem nephelometers, lidar technology allows for measurements under unmodified ambient atmospheric conditions without drying ambient aerosols, Actual aerosol properties are not as affected when measured this way (Lv et al., 2017; Bedoya-Velásquez et al., 2018). The lidar also provides an opportunity to study the vertical characterization of aerosol hygroscopicity. Many ground-based Raman lidar systems have been operated around the world for measuring both atmospheric water vapor and aerosol profiles at higher spatial and temporal resolutions (Leblanc et al., 2012; Froidevaux et al., 2013; Wang et al., 2015; Bedoya-Velásquez et al., 2018). These measurements are useful for examining the effects of aerosol hygroscopic growth on pollution events (e.g., Y.-F. Wang et al., 2012, 2017; Su et al., 2017). Many studies on aerosol hygroscopic growth are based on the surface measurements, but few studies have investigated the vertical characterization of aerosol hygroscopicity.

Xingtai as a city with a high density of heavy industries was ranked as one of the most polluted cities in central NCP. A joint field campaign was carried out in this region in the

summer of 2016. Some studies based on this campaign have been done for understanding the causes and evolution of pollution events in this region (Y.-Y. Wang et al., 2018; Zhang et al., 2018). These studies have shown that aerosols in Xintai are highly aged and internally mixed due to strong secondary formation. The goal of this study is to further investigate how aerosol hygroscopic growth affects haze events and what are the controlling factors by combining surface and vertical measurements of aerosol optical, physical, and chemical properties.

The following section describes the instruments and methodology. Section 3 presents the results and discussion. Section 4 provides a brief summary of the study.

## 2. Instruments and methodology

### 2.1 Instruments

A Raman lidar was used to analyze the relationship between atmospheric water vapor content and $PM_1$ or $PM_{2.5}$ mass concentrations, and to explore the atmospheric aerosol hygroscopic growth effect on haze events. The lidar is an automated system that retrieves atmospheric water vapor mixing ratios ($W$) and aerosol optical property profiles (aerosol extinction and backscattering coefficients, Ångström exponent (AE), and the depolarization ratio) throughout the day. The system employs a pulsed neodymium-doped yttrium aluminum garnet laser as a light source and emits three laser beams simultaneously at 355, 532, and 1064 nm with a time resolution of 15 min and a range resolution of 7.5 m based on its original factory settings. The lidar sends 5,000 laser beams in the first four minutes and ten seconds of the 15-minute cycle, then the mean value of the received 5,000 signals are stored as the signal profile

to enhance the signal-to-noise ratio. When a laser beam is sent into the atmosphere, the received

backscattering signal generally includes Mie scattering by aerosols, Rayleigh scattering by

atmospheric molecules, and Raman scattering caused by the rotation and vibration of the

molecules. The size of many molecules and atoms in the atmosphere are typically much smaller

than the wavelength of the laser, so Rayleigh scattering occurs when they interact (Strutt, 1871).

Mie scattering describes the interaction between large particles (mainly atmospheric aerosols)

and laser beams. As for the optical receiving unit of this lidar system, optical fiber (OF),

dichroic beam splitter (DBS), and ultra-narrowband filters following an ultraviolet telescope

divides atmospheric Mie scattering signals and vibrational Raman scattering signals from $H_2O$

and $N_2$ molecules (at 355, 386, and 407 nm, respectively). Atmospheric Mie scattering signals

at 532 and 1064 nm are divided by OF, DBS and ultra-narrowband filters after a visible infrared

telescope. Based on the perpendicular and parallel components at 532 nm received by the lidar

system, the aerosol depolarization ratio, a parameter that measures the shapes of aerosols, can

be calculated. In general, the more irregular the particle shape, the larger the value of the

depolarization ratio (Chen et al., 2002; Baars et al., 2016). The AE can also be calculated using

lidar signals at 532 and 1064 nm, which is inversely related to the average size of the aerosols

(Ångström, 1964; Tiwari et al., 2016).

Co-located radiosondes were launched twice a day, i.e., at ~0715 and ~1915 Beijing Time

(BJT), during the field campaign. The GTS1 detector collected profiles of atmospheric RH,

temperature, and pressure at a resolution of 1.0%, 0.1°C, and 0.1 hPa, respectively. The

radiosonde ascension velocity was typically ~5–6 m s$^{-1}$.

A co-located Doppler lidar system (TWP3-M) was also in operation at Xingtai. This

system emits electromagnetic beams in different directions to the upper air, then directly

receives the backscattering signals after those beams interact with atmospheric turbulence.

Based on the Doppler effect, this system can derive time series of horizontal wind velocity and

direction at a time resolution of 5 min and a range resolution of 60 m below 1 km and 120 m

above 1 km. The root-mean-square errors (RMSEs) of the Doppler lidar-retrieved wind speed

and direction are typically $\leq 1.5$ m s$^{-1}$ and $\leq 10^{o}$, respectively. The maximum and minimum

detection distances of this system are 3–5 km and 0.1 km, respectively.

A GrayWolf six-channel handheld particle/mass meter (model PC-3016A) was used to

directly monitor the total mass concentrations of PM$_{2.5}$ and PM$_1$ in the actual atmosphere (Yan

et al., 2017). The minimum detection particle size is 0.3 μm, the counting efficiency for 0.3-

11      μm particles is 50%, and 100% for particle sizes greater than 0.45 μm. The non-refractory PM$_1$

(NR-PM$_1$) chemical components including organics, sulfate, nitrate, ammonium, and chloride

were measured in situ by an aerodyne quadrupole aerosol chemical speciation monitor (ACSM)

at a time resolution of five minutes. Detailed information about the operation of the ACSM and

its application in this campaign can be found elsewhere (Zhang et al., 2018). Briefly, aerosols

with vacuum aerodynamic diameters of ~40–1000 nm are sampled into the ACSM through a

100-mm critical orifice mounted at the inlet of an aerodynamic lens. The particles are then

directed onto a resistively heated surface (~600$^{o}$C) where NR-PM$_1$ components are flash

vaporized and ionized by a 70-eV electron impact. The ions are then analyzed by a commercial

quadrupole mass spectrometer. Mass spectra are the raw data collected by the ACSM, and

standard analysis software offered by Aerodyne Inc. is provided to derive mass concentrations

of each chemical component. In this study, the ACSM was calibrated with pure ammonium

nitrate following the procedure detailed by Ng et al. (2011) to determine its ionization

efficiency. The aerosol aerodynamic particle size was determined by an aerodynamic lens. The

uncertainties of ACSM-derived quantities are insignificant (Ng et al. 2011).

The aerosol GF probability distribution function (GF-PDF) at RH = 85% was measured

by an in situ H-TDMA. The H-TDMA system mainly consists of a Nafion dryer, a bipolar

neutralizer, two DMAs, a CPC, and a Nafion humidifier. The first DMA is used to select

monodispersed aerosols with a set mobility size (40, 80, 110, 150, and 200 nm in this study)

after the sample is dried and neutralized by the Nafion dryer and the bipolar neutralizer. The

selected particles are then humidified when passing through a Nafion humidifier with

controlled RH (85%). The second DMA and the CPC are responsible for measuring the number

size distribution of the humidified particles. Finally, the TDMA-fit algorithm is used to retrieve

GF-PDF (Stolzenburg and McMurry, 2008). Uncertainties of these retrieved parameters are

insignificant. More detailed descriptions about the H-TDMA system are given by Tan et al.

(2013) and Y.-Y. Wang et al. (2017, 2018). All data are reported in BJT in this study.

**2.2 Methodology**

**2.2.1 Water vapor retrieval**

Using the ratio of the Raman signals of $H_2O$ ( $P_H$ ) and $N_2$ ( $P_N$ ), $W$ is calculated as follows

(Melfi, 1972; Leblanc et al., 2012; Su et al., 2017):

$$W(z) = C_W \Delta q \frac{P_H(z)}{P_N(z)} \quad , \tag{1}$$

$$\Delta q = \frac{\exp[-\int_0^z (\alpha_N^m + \alpha_N^p) dz]}{\exp[-\int_0^z (\alpha_H^m + \alpha_H^p) dz]} \quad , \qquad (2)$$

where $C_W$ is the Raman lidar calibration constant which can be calculated using co-located

radiosonde data (Melfi, 1972; Sherlock et al., 1999). The parameters $\alpha_N^m$ and $\alpha_H^m$ are the

molecular extinction coefficients at 386 and 407 nm, respectively. These can also be calculated

using temperature and pressure profiles from radiosonde measurements (Bucholtz, 1995). The

parameters $\alpha_N^p$ and $\alpha_H^p$ are the aerosol extinction coefficients (AECs) at 386 and 407 nm,

respectively. Here, we use the Fernald method to retrieve AECs (Fernald et al., 1972; Fernald,

1984), which is an analytic solution to the following basic lidar equation for Mie scattering:

$$P_s(z) = ECZ^{-2}[\beta_1(z) + \beta_2(z)]T_1^2(z)T_2^2(z), \qquad (3)$$

where $P_s(z)$ is the return signal, $E$ is the energy emitted by the laser, $C$ is the calibration

constant of the lidar system, and $\beta_1(z)$ and $\beta_2(z)$ are the backscattering cross-sections of

atmospheric aerosols and molecules at altitude $z$, respectively. $T_1(z)$ and $T_2(z)$ are the

transmittances of aerosols and air molecules at height $z$. Note that during the daytime, the height

of the retrieved $W$ profile is limited because the Raman signal is affected by radiation (Tobin

et al., 2012).

We can also calculate the vertical distribution of RH based on the vertical profile of $W$

retrieved from Raman lidar measurements and the temperature and pressure profiles provided

by radiosonde data. The following equations are used to retrieve the RH profile:

$$RH(z) = \left[\frac{e(z)}{e_s(z)}\right] \times 100\% , \qquad (4)$$

$$e(z) = \frac{W(z)p(z)}{0.622 + W(z)} , \qquad (5)$$

$$e_s(z) = 6.1078 \exp\left[\frac{17.13[T(z) - 273.16]}{T(z) - 38}\right], \tag{6}$$

where $e(z)$ and $e_s(z)$ are the vertical profiles of water vapor pressure (in hPa) and

saturation vapor pressure (in hPa) at a certain temperature, respectively, $W(z)$ is the $W$ profile

obtained from the Raman lidar, $p(z)$ is the pressure profile (in hPa), and $T(z)$ is the

temperature profile (in K) provided by radiosonde data.

To assess the accuracy of the retrieval algorithm, Raman lidar- and radiosonde-derived $W$

and RH profiles at 0515 BJT on 24 May 2016 and their differences are shown in Fig. 1. The $W$

profiles agree reasonably well with an absolute error between them less than 0.5 g kg$^{-1}$.

Absolute errors between Raman lidar- and radiosonde-derived RH profiles are generally less

than 5%. The same inversion results for a relatively wet case on 23 May 2016 are given in Fig.

2. In general, large absolute errors tend to occur at the inflection points. Figures 1 and 2 suggest

that the retrieval algorithm can produce reasonable results.

**2.2.2 Selection of aerosol hygroscopic cases and their optical properties**

How aerosol hygroscopic growth cases were chosen is described here. First, atmospheric

mixing conditions were examined using radiosonde-based vertical potential temperature ($\theta$)

and $W$ profiles. Cases with near-constant values of $\theta$ and $W$ in the analyzed layer (variations

less than 2°C and 2 g kg$^{-1}$, respectively) represent good atmospheric mixing conditions

(Granados-Muñoz et al., 2015). Then aerosol backscattering coefficient profiles at 532 nm were

calculated using the Fernald method (see details in section 2.2.1).

A simultaneous increase in atmospheric RH and the aerosol backscattering coefficient is

also needed, which might indicate aerosol hygroscopic growth (Bedoya-Velásquez et al., 2018).

Based on the above criteria, individual cases with the same ambient humidity and different

pollution conditions were selected for studying the influence of aerosol hygroscopicity on haze

events. Aerosol hygroscopic properties of the selected cases were investigated in terms of the

hygroscopic enhancement factor for the aerosol backscattering coefficient defined as follows:

$$f_\beta\left(RH,\lambda\right)=\frac{\beta\left(RH,\lambda\right)}{\beta(RH_{ref},\lambda)} \quad , \qquad\qquad (7)$$

where $\beta\left(RH,\lambda\right)$ and $\beta\left(RH_{ref},\lambda\right)$ represent aerosol backscattering coefficients at a

certain RH value and at a reference RH value, respectively, at wavelength $\lambda$. In this study,

we selected $RH_{ref}=80\%$ which is the lowest RH in the layer.

Finally, a relationship between $f_\beta(RH)$ and RH was established. The most commonly

used equations are the single-parameter fit equation (e.g., Hänel, 1980; Kotchenruther and

Hobbs, 1998; Gassó et al., 2000) and the dual-parameter fit equation (e.g., Hänel, 1980; Carrico,

2003; Zieger et al., 2011). The single-parameter fit equation introduced by Hänel (1976) is

$$f_\beta\left(RH\right)=\left(\frac{1-RH}{1-RH_{ref}}\right)^{-\gamma}, \qquad\qquad (8)$$

where $\gamma$ in an empirical parameter. Larger $\gamma$ values in this formulation denote a stronger

hygroscopic growth.

The dual-parameter fit equation is (Fernández et al., 2015)

$$f_\beta\left(RH\right)=a(1-RH)^{-b}. \qquad\qquad (9)$$

The single- and dual-parameter fit equations are similar, but with an additional scale factor

parameter, $a$, in the case of the dual-parameter fit equation. The parameter $b$ is also an

empirical parameter with larger values of $b$ indicating particles with stronger hygroscopicities.

In this study, both parameterized equations are used to verify the consistency of the results. The

equation that fits the measurement data best is selected.

### 2.2.3 Calculation of aerosol acidity

Aerosol acidity is associated with aerosol hygroscopic growth (e.g. Sun et al., 2009; Fu et al., 2015; Zhang et al., 2015; Lv et al., 2017). When atmospheric aerosols are acidic, they have stronger hygroscopicities than when in their neutralized forms (Zhang et al., 2015). The swelling of aerosols due to hygroscopic growth enhances their ability to scatter solar radiation. We examined acidity by comparing the measured $NH_4^+$ mass concentration with the needed amount to fully neutralize sulfate, nitrate, and chloride ions ($NH_{4\ predicted}^+$) detected by the ACSM (Sun et al., 2009; Zhang et al., 2015; Lv et al., 2017):

$$NH_{4\ predicted}^+ = (2 \times SO_4^{2-} / 96 + NO_3^- / 62 + Cl^- / 35.5) \times 18, \tag{10}$$

where $SO_4^{2-}$, $NO_3^-$, and $Cl^-$ represent the mass concentrations (in μg m$^{-3}$) of the three species measured by the ACSM. The molecular weights of $SO_4^{2-}$, $NO_3^-$, $Cl^-$, and $NH_4^+$ are 96, 62, 35.5, and 18, respectively. Aerosols are considered "more acidic" if the measured $NH_4^+$ mass concentration is significantly lower than that of $NH_{4\ predicted}^+$. Aerosols are considered "bulk neutralized" if the two values are similar (Zhang et al., 2007; Sun et al., 2009; Zhang et al., 2015; Lv et al., 2017).

The acidity of aerosols can be quantified by a parameter called the acid value (AV) (Zhang et al., 2007):

$$AV = (2 \times SO_4^{2-} / 96 + NO_3^- / 62 + Cl^- / 35.5) / (NH_4^+ / 18). \tag{11}$$

The chemical formula and numbers after the equal sign have the same meanings as in Eq. (10).

Aerosols are considered "bulk neutralized" if $AV = 1$ and "strongly acidic" if $AV > 1.25$. When

$AV = 1.25$, 50% of the total sulfate ions in the atmosphere consist of $NH_4HSO_4$, and the other

50% consist of $(NH_4)_2SO_4$.

**2.2.4 Aerosol chemical ion-pairing scheme**

The magnitude of $f(RH)$ is correlated with the inorganic mass fraction (Zieger et al., 2014).

However, GFs differ with different inorganic salts. To examine the mass fractions of neutral

inorganic salts, ACSM measurements were used to calculate their mass concentrations and

volume fractions (Gysel et al., 2007). This approach is based on the ion-pairing scheme

introduced by Reilly and Wood (1969). The ACSM mainly measures the mass concentrations

of $SO_4^{2-}$, $NO_3^-$, $NH_4^+$, $Cl^-$, and organics. The chlorine ion was not considered here because

its concentration is low. The aerosol chemical ion combination scheme is given by the

following equations:

$$
\begin{aligned}
n_{NH_4NO_3} &= n_{NO_3^-} \\
n_{NH_4HSO_4} &= \min(2n_{SO_4^{2-}} - n_{NH_4^+} + n_{NO_3^-}, n_{NH_4^+} - n_{NO_3^-}) \\
n_{(NH_4)_2SO_4} &= \max(n_{NH_4^+} - n_{NO_3^-} - n_{SO_4^{2-}}, 0) \\
n_{H_2SO_4} &= \max(0, n_{SO_4^{2-}} - n_{NH_4^+} + n_{NO_3^-}) \\
n_{HNO_3} &= 0
\end{aligned}
\qquad , \qquad (12)
$$

where $n$ donates the mole numbers, and "min" and "max" are minimum and maximum values

(Gysel et al., 2007). The volume fractions of inorganic salts can be calculated based on the ion

combination scheme and the parameters in Table 1. Furthermore, for a multicomponent particle,

the Zdanovskii-Stocks-Robinson mixing rule (Zdanovskii, 1948; Stokes and Robinson, 1966)

can be applied to calculate the hygroscopicity parameter $\kappa$:

$$
\kappa = \sum_i \varepsilon_i \kappa_i \quad , \qquad (13)
$$

where $\kappa_i$ is the hygroscopicity parameter of each individual component. The parameter $\varepsilon_i$ is the volume fraction of each component.

**3.  Results and discussion**

**3.1 Observations of $W$ and mass concentrations of PM$_1$ and PM$_{2.5}$**

Figure 3a shows the time series of the lidar-derived $W$ at Xingtai from 19–31 May 2016. The height of the retrieved $W$ profile was limited because of solar radiation during the daytime (e.g., Tobin et al., 2012). Overall, $W$ is generally less than 6 g kg$^{-1}$ between 0.3–4 km with a strong daily variability during the period analyzed. Figures 3b and 3c show the simultaneous time series of the surface mass concentrations of PM$_1$ and PM$_{2.5}$, and $W$ and RH, respectively. The variabilities in PM$_1$ and PM$_{2.5}$ mass concentrations are strongly coupled with that in $W$ at the surface and in the lower atmospheric layer. Others have also found the same relationship between $W$ in the lower atmospheric layer and the surface mass concentration of PM$_{2.5}$ (e.g., Y.-F. Wang et al., 2012, 2017; Su et al., 2017). Su et al. (2017) suggested that this is due to the aerosol hygroscopic growth effect. The aerosol hygroscopicity is related to aerosol chemical composition over the North China Plain (Zou et al., 2018). Figure 3d shows simultaneous mass fractions of the chemical species comprising PM$_1$. As $W$ in the lower atmospheric layer and the surface mass concentrations of PM$_1$ and PM$_{2.5}$ increases, the proportion of organic aerosols decreases (highlighted as shaded grey areas in Fig. 3), suggesting that the proportion of hygroscopic aerosols increased. This shows that strong aerosol hygroscopicity may aggravate air pollution conditions over Xingtai.

Two instances when this relationship was not seen (highlighted as shaded grey areas in

Fig. 3) are shown by the black triangles in Fig. 3d and marked with grey lines across Fig. 3. In

the evening of 21 May 2016 (the leftmost triangle and grey line), $W$ and the mass fractions of

organics are comparable to those in the evening of 23 May (the rightmost triangle and grey line

in Fig. 3). However, the mass concentrations of $PM_1$ and $PM_{2.5}$ at that time indicated by the

leftmost grey line (in the evening of 21 May 2016) are significantly less than that in the evening

of 23 May (indicated by the rightmost grey line). Su et al. (2017) and Y.-F. Wang et al. (2012,

2017) have studied the relationship between atmospheric water vapor and haze events over

Beijing and Xi'an, respectively, using Raman lidar measurements. Their analyses showed a

positive correlation between $W$, and $PM_{10}$ and $PM_{2.5}$ mass concentrations, but they ignored

some unexpected cases behind this positive correlation. The two unexpected cases that

occurred on 21 May 2016 (Case I) and 23 May 2016 (Case II) were selected for further study.

**3.2 Cases studies of aerosol hygroscopic growth**

**3.2.1 Lidar-estimated hygroscopic measurements**

Two cases were selected: one on 21 May 2016 (Case I) and the other on 23 May 2016

(Case II) at the closest time of the radiosonde launch time at 1915 BJT. Figure 4 shows the

vertical distributions of $W$, $\theta$, the aerosol backscattering coefficient at 532 nm ($\beta_{532}$), the

backscatter-related Ångström exponent (AE) based on measurements at 532 and 1064 nm, and

the particle linear depolarization ratio at 532 nm for Case I and Case II. The altitude ranges are

1642.5–1905.5 m for Case I and 1680.0–2130.0 m for Case II. $W$ and $\theta$ calculated from

radiosonde-measured temperature and RH profiles were used to examine the atmospheric

mixing conditions in the individual layers. Table 2 lists the gradients of the variables within

each layer. The gradient in $W$ changes little within the layer of interest, decreasing

monotonically with altitude at a rate of -0.34 g kg$^{-1}$ km$^{-1}$ and -1.42 g kg$^{-1}$ km$^{-1}$ for Case I and

Case II, respectively. The gradient in $\theta$ shows a monotonic increase within the layers of interest

(0.27$^{o}$C km$^{-1}$ for Case I and 0.96$^{o}$C km$^{-1}$ for Case II). Overall, $W$ and $\theta$ variations are less than

2 g kg$^{-1}$ and 2$^{o}$C, respectively, showing that good mixing atmospheric conditions were present

in both cases (Granados-Muñoz et al., 2015). This confirms that aerosols within the analyzed

layer of each case were well mixed.

Figure 4 shows the time series of the horizontal wind velocity and direction retrieved from

the co-located Doppler lidar system. From 1830–2030 BJT, Case I (Fig. 4c) and Case II (Fig.

4d) winds within their respective layers are mainly from the north and northwest, respectively,

and have relatively low speeds (< 5 m s$^{-1}$, Fig. 4a and 4b). This suggests that the aerosols in

each case were transported into their respective layers at low speeds from almost the same

direction. In other words, there is no change in the aerosol type of both cases within the region

of interest.

The RH and $\beta_{532}$ simultaneously increase with altitude in the Case I (Fig. 5c and 5d) and

Case II (Fig. 5i and 5j) layers of interest. The AE and depolarization ratio were retrieved in

order to differentiate the fine/coarse mode predominance and shape of the aerosols (Fig. 5e, f,

k, and l). A decrease in AE and the depolarization ratio suggests that there is an increase in the

predominance of coarse-mode particles and an increase in the sphericity of particles due to

water uptake, respectively (Granados-Muñoz et al., 2015; Lv et al,. 2017; Bedoya-Velásquez

et al., 2018).

Based on the $\beta_{532}$ and RH profiles retrieved from Raman lidar measurements, the enhancement factor for the backscattering coefficient at 532 nm, $f_{\beta}(RH)$, is calculated for both cases using Eq. (7). The reference RH value was set to 80% in this study, the lowest RH recorded in the layers of interest of both cases. This study applies the single-parameter Hänel model [Eq. (8)] and the dual-parameter Kasten model [Eq. (9)]. Table 3 lists the parameterized results of each model for each case, and Fig. 6 shows the best-fit lines. The $f_{\beta}(RH)$ for Case II is greater than that for Case I. $\beta_{532}$ increases by a factor of 1.094 (Case I) and 1.794 (Case II) as RH changes from 80% to 91%. The magnitudes of $f_{\beta}(85\%)$ for Case I and Case II are 1.0283 and 1.0770, respectively. The $b$ value from the Kasten parameterization is much larger in Case II (0.9346) than in Case I (0.1000), and the γ value from Hänel parameterization for Case II (0.6538) is also much larger than that for Case I (0.09895). Chen et al. (2014) studied the aerosol hygroscopicity parameter derived from light-scattering enhancement factor [$f(RH)$] measurements made in the NCP and showed that $f(RH)$ for polluted cases is higher than that for clean periods at a specific RH. This is consistent with the results of this study where the mass concentrations of $PM_1$ and $PM_{2.5}$ during Case II (69.36 μg m$^{-3}$ for $PM_1$ and 94.88 μg m$^{-3}$ for $PM_{2.5}$) were greater than those during Case I (34.08 μg m$^{-3}$ for $PM_1$ and 45.00 μg m$^{-3}$ for $PM_{2.5}$). An observational study of the influence of aerosol hygroscopic growth on the scattering coefficient at a rural area near Beijing also demonstrated that aerosols had relatively strong water-absorbing properties during urban pollution periods (Pan et al., 2009).

**3.2.2 The influences of chemical composition inferred from ACSM measurements**

Liu et al. (2014) have pointed out that inorganics are the primary aerosol component contributing to aerosol hygroscopicity especially in the size range of 150–1000 nm. The acidity

of aerosols is a key parameter affecting aerosol hygroscopic growth (Sun et al., 2009; Lv et al.,

2017). The dominant form of inorganics can be examined by comparing measured $NH_4^+$ and

predicted $NH_4^+$ (Lv et al., 2017; see section 2.2.3 for details).

Figure 7 shows the relationships between ACSM-measured $NH_4^+$ and predicted $NH_4^+$

based on PM$_1$ chemical species information collected during the full day of each case. The

slopes of the linear regression best-fit lines are 0.72 and 0.68 on 21 May 2016 (Case I) and 23

May 2016 (Case II), respectively. The RMSEs of the liner regression best-fit lines are 0.63 and

0.48 on 21 May 2016 and 23 May 2016, respectively. The parameter $AV$ for Case I is 1.35 and

for Case II is 1.50. These values suggest that there was insufficient NH$_3$ in the atmosphere to

neutralize H$_2$SO$_4$, HNO$_3$, and HCl in each case and that the dominant form of inorganics was

NH$_4$NO$_3$, NH$_4$HSO$_4$, and (NH$_4$)$_2$SO$_4$. The acidity of aerosols in Case II is greater than that in

Case I, suggesting that aerosols in Case II were more hygroscopic than those in Case I. This is

consistent with the results presented in section 3.2.1.

A hygroscopicity parameter, $\kappa$, was developed by Petters and Kreidenweis (2007). $\kappa$ can

be calculated using the chemical composition information from Eq. (13) (Gysel et al., 2007;

Y.-C. Liu et al., 2016; see section 2.3.4). To further confirm the effect of aerosol hygroscopic

growth on haze events, $\kappa$ is computed for each case based on the dominant form of the

inorganics mentioned above.

Figure 8 shows the chemical species obtained from ground-based ACSM measurements

of PM$_1$ around the times of the cases. In Case I (Fig. 8a), PM$_1$ was mainly made up of organic

particles (39%) and sulfate (39%), followed by nitrate (8%), ammonium (13%), and chloride

(1%). In Case II (Fig. 8b), $PM_1$ was made up of 37% organics, 25% sulfate, 22% nitrate, 12%

ammonium, and 1% chloride. Based on the aerosol chemical ion-pairing scheme introduced in

Section 2.2.4 and the aerosol properties shown in Table 1, chloride and organics were neglected

because of their relatively small contents and comparatively low hygroscopicities (Gysel et al.,

2007; Petters and Kreidenweis, 2013). Table 4 lists the mass concentrations and volume

fractions of $NH_4NO_3$, $NH_4HSO_4$, and $(NH_4)_2SO_4$ for each case as well as $\kappa$ computed using

Eq. (13). The mass concentration of $H_2SO_4$ is equal to zero. Liu et al. (2014) have shown that

$\kappa$ for $NH_4NO_3$, $NH_4HSO_4$, and $(NH_4)_2SO_4$ is equal to 0.68, 0.56, and 0.60, respectively. The

parameter $\kappa$ for Case I (0.557) is less than that for Case II (0.610). This suggests that the aerosol

hygroscopicity for Case II was higher than that for Case I. It also suggests that under the same

ambient RH conditions, the nitrate content in aerosols can cause differences in the

hygroscopicity of aerosols.

**3.2.3 Comparison with H-TDMA measurements**

In the last decade, many studies have compared remotely sensed and in situ aerosol

scattering enhancement factor measurements using a humidified tandem nephelometer and

have shown positive results (Zieger et al., 2011, 2012; Sheridan et al., 2012; Tesche et al., 2014;

Lv et al., 2017). The H-TDMA is also a reliable instrument for measuring the aerosol diameter

hygroscopicity due to water uptake (Liu et al., 1978). Aerosol GFs observed by the ground-

based H-TDMA at times nearest to the times of each case are examined next.

Based on H-TDMA-derived aerosol GFs at an RH level of 85% for different particle sizes

(40, 80, 110, 150, and 200 nm), GFs for different aerosol sizes in both cases were extrapolated

to different RH levels using Eq. (3) from\(2009) who used the κ model introduced by Petters

and Kreidenwies (2007). Figure 9 shows that Case II aerosol GFs at each RH level (80–91%)

are higher than those of Case I. Although the $f_\beta(RH)$ and GF are completely different

parameters for calculating the hygroscopicity of aerosols and are difficult to compare

quantitatively, the H-TDMA results offer a sense of confidence that aerosol hygroscopicity has

an important influence on the formation of heavy haze.

In general, both the lidar-estimated aerosol backscattering hygroscopic enhancement

factor and the ACSM and H-TDMA measurements support the proposed hypothesis that the

different hygroscopic properties of aerosols are mainly responsible for the strong coupling

between the variability in $PM_1$ and $PM_{2.5}$ mass concentrations and the variability in $W$.

**4. Conclusions**

During late May 2016, the water vapor mixing ratio in the 0.3–4 km layer over Xingtai

was generally less than 6 g $kg^{-1}$ with a strong daily variability. Overall, the simultaneous

temporal changes in the mass concentrations of $PM_1$ and $PM_{2.5}$ were strongly associated with

that of the atmospheric water vapor content due to the hygroscopicity of aerosols. Two cases

where this relationship was not seen were identified and further examined. Case I represents a

relatively clean case, and Case II represents a polluted case. The lidar-estimated aerosol

backscattering coefficient hygroscopic enhancement factor [$f_\beta(RH)$] for Case II is greater

than that for Case I. The $\gamma$ and $b$ values from the Hänel and Kasten parameterizations,

respectively, for Case II were larger than those for Case I. A key parameter affecting the

hygroscopicity of aerosols, namely, the acid value ($AV$), was examined by comparing measured

$NH_4^+$ and predicted $NH_4^+$ based on data obtained by the ACSM. The *AV* for Case I (1.35) was less than that for Case II (1.50) and the main form of inorganics was $NH_4NO_3$, $NH_4HSO_4$, and $(NH_4)_2SO_4$. The aerosol chemical composition determined by the ACSM showed that the aerosol hygroscopicity parameter $\kappa$ for Case II (0.610) was greater than that for Case I (0.577) due to the greater mass fraction of nitrate salt. Based on H-TDMA measurements, model results showed that the aerosol size hygroscopic growth factor (GF) in each particle size category (40, 80, 110, 150, and 200 nm) for Case II was greater than that for Case I.

The $f_\beta(RH)$, GF, *AV*, and $\kappa$ are completely different quantities for calculating the hygroscopicity of aerosols and are difficult to compare quantitatively. The lidar-estimated $f_\beta(RH)$ and ACSM and H-TDMA measurements show that the hygroscopic growth of aerosols has a strong influence on the process of air pollution. Under the same atmospheric relative humidity conditions, the stronger the hygroscopicity of aerosols, the more likely they cause severe air pollution. The mass fraction of the nitrate ion in aerosols was one of the main factors that determined the hygroscopic ability of aerosols in the study area (Xingtai). These findings not only reveal why haze events in Xintai can be severe, but they also provide scientific evidence that may be used to persuade the local government to prevent and control environmental contamination in this heavily polluted part of China.

**Author contributions**

ZL and JC determined the main goal of this study. JC carried it out, analyzed the data and prepared the paper with contributions from all co-authors. YW provided technical guidance for related instruments.

**Data availability**

Data used in this study are available from the first author upon request (jchen@mail.bnu.edu.cn).

**Competing interests**

The authors declare that they have no conflict of interest.

**Special issue statement**

This article is part of the special issue "Regional transport and transformation of air pollution in eastern China". It is not associated with a conference.

**Acknowledgements**

This work was supported by the National Key R&D Program of China (2017YFC1501702), the National Science Foundation of China (91544217), and the U.S. National Science Foundation (AGS1534670).

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

**Table 1. Aerosol properties of selected compounds used for the calculation of the hygroscopicity**

**parameter $\kappa$, i.e., the density ($\rho_i$) and ($\kappa_i$) of each compound.**

| species | NH$_4$NO$_3$ | NH$_4$HSO$_4$ | (NH$_4$)$_2$SO$_4$ | H$_2$SO$_4$ |
|---------|--------------|---------------|---------------------|-------------|
| density[a] | 1.725 | 1.78 | 1.76 | 1.83 |
| $\kappa$ [b] | 0.68 | 0.56 | 0.52 | 0.91 |

(a) Tang and Munkelwitz (1994); Carrico et al. (2010);

(b) Fountoukis and Nenes (2007); Carrico et al. (2010); Liu et al. (2014).

**Table 2. Range of values and gradient values over the analyzed layer for the water vapor mixing ratio ($W$), the potential temperature ($\theta$), the backscattering coefficient at 532 nm ($\beta_{532}$), the Ångström exponent [AE (532–1064 nm)], and the depolarization ratio at 532 nm for Cases I and II.**

| | Case I | | Gradient | Case II | | Gradient |
|---|---|---|---|---|---|---|
| | Range | | $(km^{-1})$ | Range | | $(km^{-1})$ |
| Altitude (m) | 1642.5 | 1905.0 | — | 1680.0 | 2130.0 | — |
| $W$ (g kg$^{-1}$) | 7.65 | 7.56 | -0.34 | 6.42 | 5.78 | -1.42 |
| $\theta$ (℃) | 26.93 | 27.00 | 0.27 | 25.18 | 25.61 | 0.96 |
| RH (%) | 80 | 91 | — | 80 | 91 | — |
| $\beta_{532nm}$ (km$^{-1}$ sr$^{-1}$) | 0.01379 | 0.01535 | — | 0.003711 | 0.006762 | — |
| AE (532–1064 nm) | 0.74 | 0.68 | -0.23 | 0.42 | 0.35 | -0.16 |
| Depolarization ratio | 0.046 | 0.044 | -0.0076 | 0.041 | 0.039 | -0.0044 |

**Table 3. The fitting parameters and $R^2$ of the fits for the Kasten and Hänel models.**

|  | Case I | | | Case II | | |
|---|---|---|---|---|---|---|
|  | a | b | $R^2$ | a | b | $R^2$ |
| Kasten model | 0.8508 | 0.1000 | 0.97 | 0.1916 | 0.9346 | 0.95 |
|  | $\gamma$ | | $R^2$ | $\gamma$ | | $R^2$ |
| Hänel model | 0.09895±0.0047 | | 0.97 | 0.6538±0.0662 | | 0.84 |

**Table 4. Calculated mass concentrations and volume fractions of $NH_4NO_3$, $NH_4HSO_4$, and $(NH_4)_2SO_4$, and the hygroscopicity parameter ($\kappa$) for Case I and Case II.**

|  | Case I | | | Case II | | |
|---|---|---|---|---|---|---|
|  | $NH_4NO_3$ | $NH_4HSO_4$ | $(NH_4)_2SO_4$ | $NH_4NO_3$ | $NH_4HSO_4$ | $(NH_4)_2SO_4$ |
| mass conc. ($\mu g\ m^{-3}$) | 3.60 | 8.31 | 8.30 | 12.2979 | 10.3795 | 3.0616 |
| volume fraction | 0.18 | 0.41 | 0.41 | 0.48 | 0.40 | 0.12 |
| $\kappa$ |  | 0.557 |  |  | 0.610 |  |

**Fig. 1. (a, c) Water vapor mixing ratio (*W*) and relative humidity (RH) profiles at 0515 BJT 24 May**

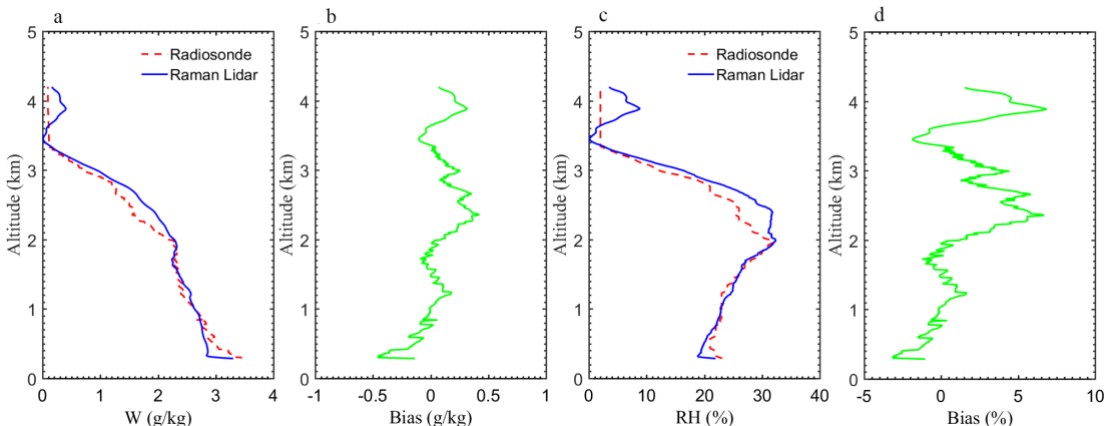

**2016 retrieved by the Raman lidar (blue line) and the radiosonde (red dashed line), respectively,**

**and (b, d) the absolute error in *W* and RH between the lidar and radiosonde retrievals (lidar minus**

**radiosonde), respectively.**

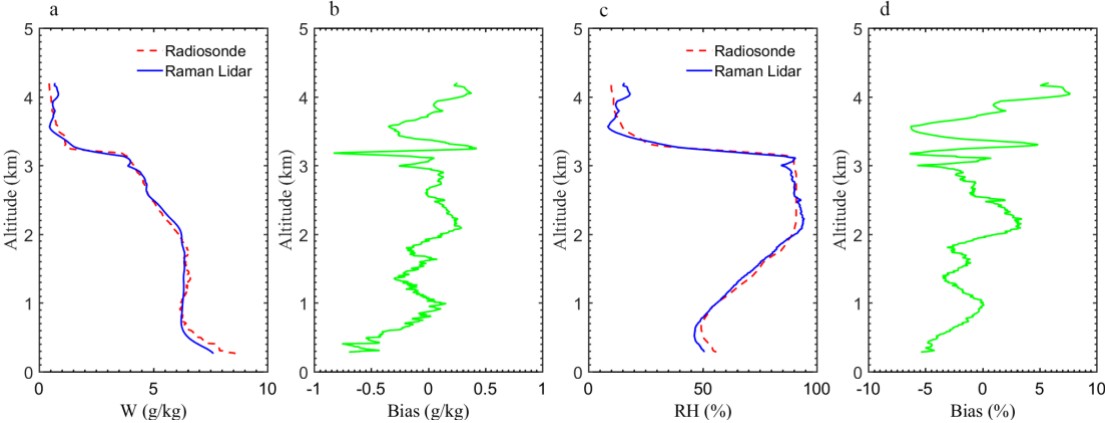

**Fig. 2. (a, c) Water vapor mixing ratio (*W*) and relative humidity (RH) profiles at 2000 BJT 23 May**

**2016 retrieved by the Raman lidar (blue line) and the radiosonde (red dashed line), respectively,**

**and (b, d) the absolute error in *W* and RH between the lidar and radiosonde retrievals (lidar minus**

**radiosonde), respectively.**

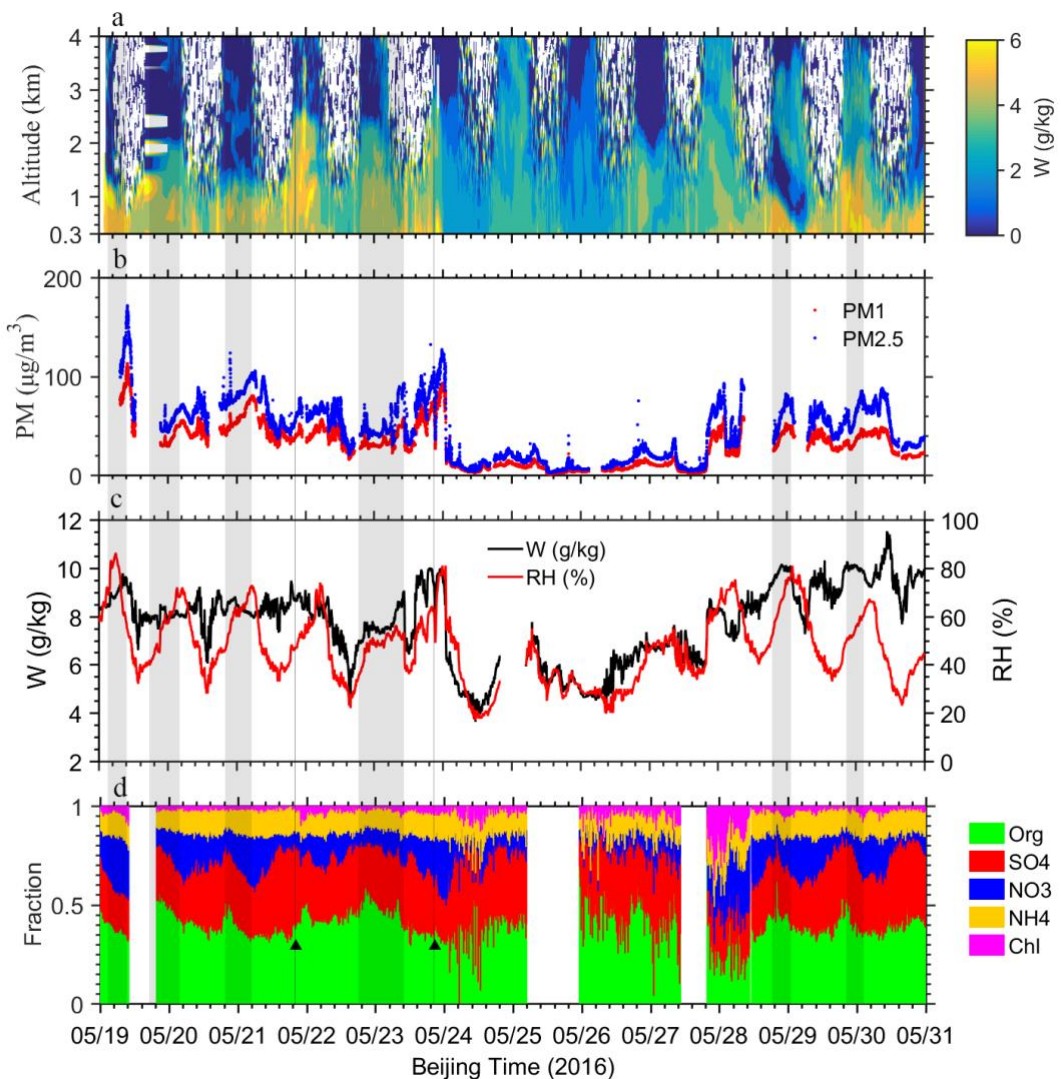

**Fig. 3.** Time series of (a) water vapor mixing ratio (*W*) profiles measured by the Raman lidar, (b)

mass concentrations of PM$_1$ (red dots) and PM$_{2.5}$ (blue dots), (c) surface *W* (black line) and relative

humidity (RH, red line), and (d) chemical species mass fractions of PM$_1$ measured by the ACSM.

Data are from 19–31 May 2016 at Xingtai. The shaded grey areas are to enhance the readability of

the article. The black triangles in (d) and grey lines in (a, b, c, d) represent the two cases chosen for

further examination. Blank parts of the data are missing due to uncontrollable factors such as

power supply.

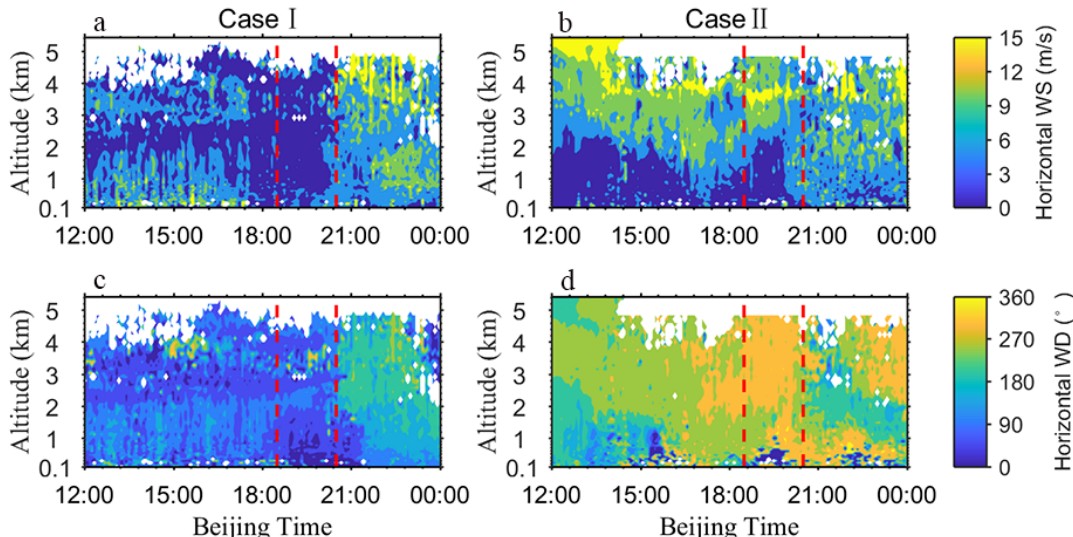

**Fig. 4. Time series of Doppler-lidar-retrieved (a, b) horizontal wind speed and (c, d) horizontal wind direction on 21 May 2016 (Case I, left-hand panels) and 23 May 2016 (Case II, right-hand panels). Red dashed lines outline the time range 1830–2030 BJT. The analyzed layers are 1642.5–1905.0 m for Case I and 1680.0–2130.0 m for Case II.**

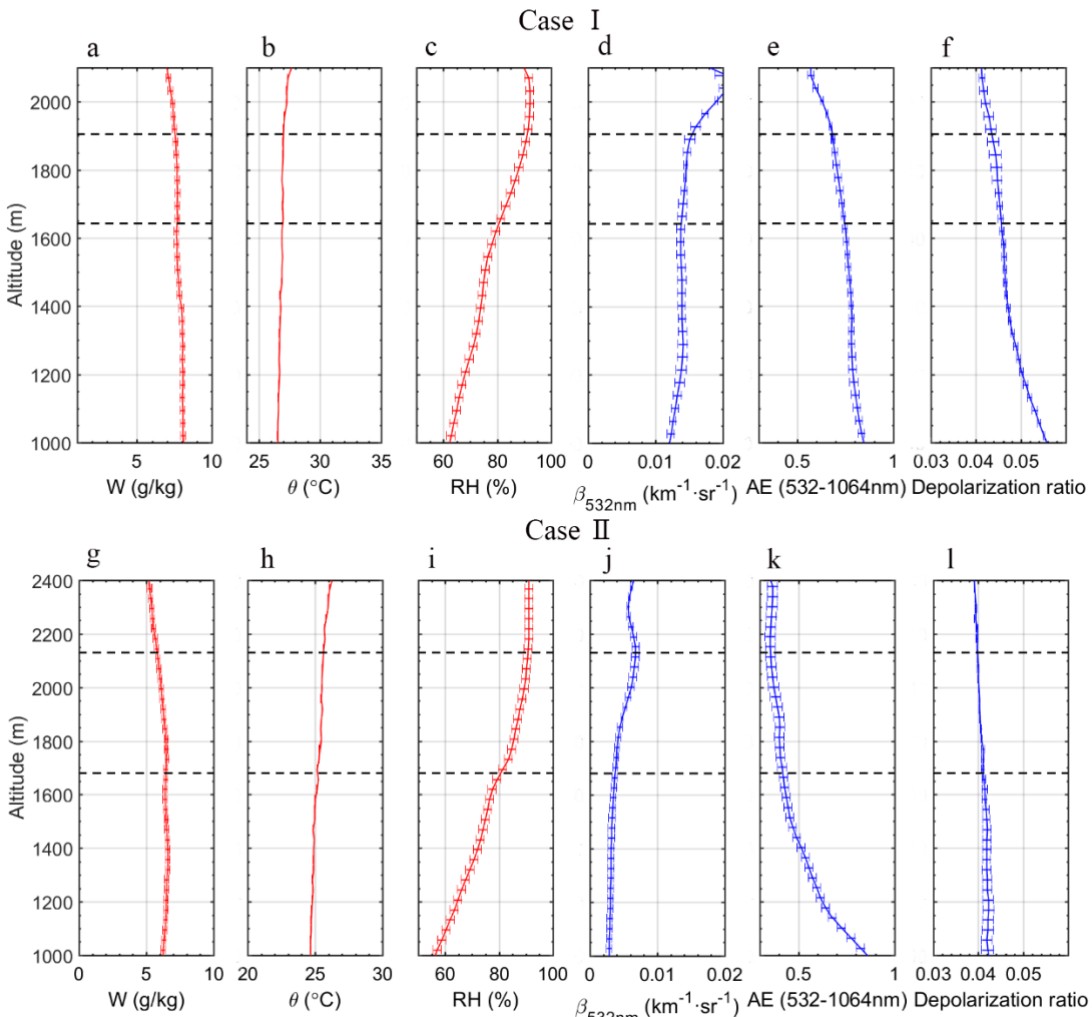

**Fig. 5. The vertical profiles of (a, g) water vapor mixing ratio ($W$), (b, h) potential temperature ($\theta$), (c, i) relative humidity (RH) calculated from radiosonde data, (d, j) backscattering coefficient at 532 nm ($\beta_{532}$), (e, k) the Ångström exponent [AE (532-1064nm)], (f, l) depolarization ratio retrieved from Raman lidar data for Case I (top panels) and Case II (bottom panels). Horizontal dashed lines show the upper and lower boundaries of the layer under analysis (1642.5–1905.0 m for Case I and 1680.0–2130.0 m for Case II). Horizontal error bars denote the uncertainty of each property.**

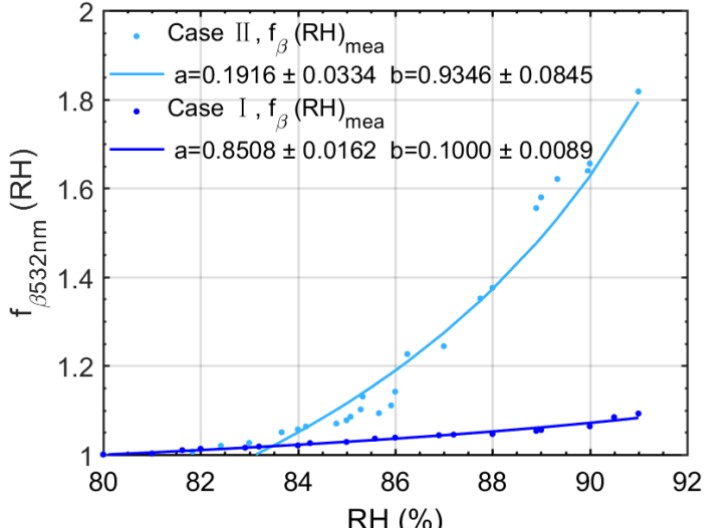

**Fig. 6.** *fβ(RH)* **at 532 nm retrieved on 21 May 2016 in the 1642.5–1905.0 m layer (Case I, dark blue points) and 23 May 2016 in the 1680.0–2130.0 m layer (Case II, light blue points). The best-fit lines through the points are shown. The reference RH is 80 %.**

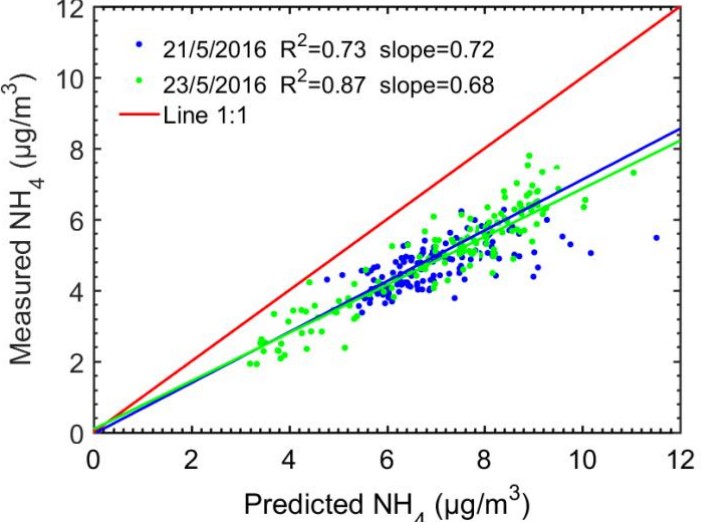

**Fig. 7. Mass concentrations of measured ammonium (NH₄) versus predicted ammonium assuming full neutralization of sulfate, nitrate and chloride on the whole day of 21 May 2016 (blue dots, Case I) and 23 May 2016 (green dots, Case II). The solid blue and green lines are the least-squares regression lines for each day. The 1:1 line is shown in red.**

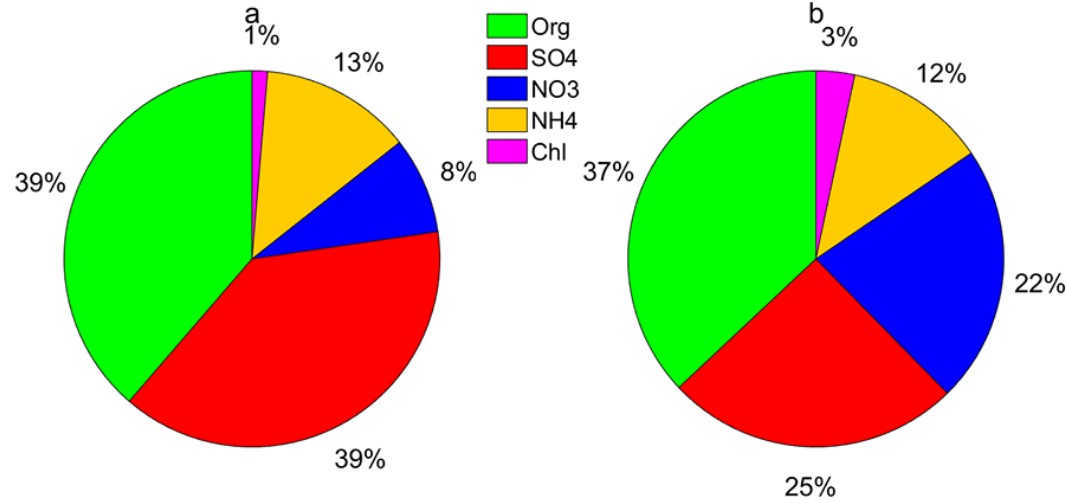

**Fig. 8. Aerosol mass fractions of PM$_1$ measured by the ACSM for (a) Case I and (b) Case II.**

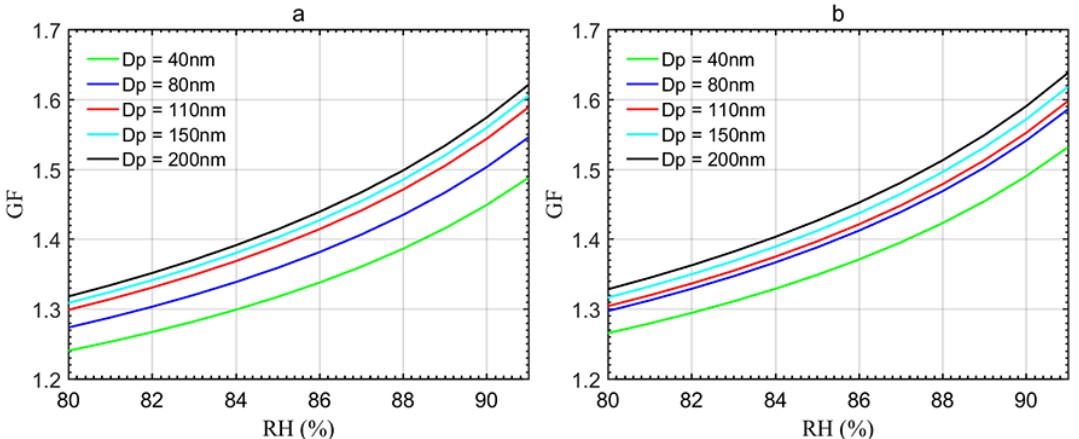

**Fig. 9. Aerosols size hygroscopic growth factor (GF) as a function of relative humidity (RH) for (a) Case I and (b) Case II. The different colors represent different particle sizes (Dp). These are the results of a model based on Eq. 3 from Gysel et al. (2009).**