# Peer review of "Aerosol hygroscopic growth, contributing factors, and impact on haze events in a severely polluted region in northern China"

_Atmospheric Chemistry and Physics, 2018_

## Referee Comment (RC1) · Anonymous Referee #1 · 14 Sep 2018

Summary:

Chen et al. report data retrieved in Xingtai, a city in the southern Hebei province of China, during the late spring of 2016, to determine to what extent (if any) aerosol hygroscopic growth contributes to severe haze events. They take advantage of data from a lidar, an aerosol chemical speciation monitor (ACSM), a hygroscopicity-tandem differential mobility analyzer (H-TDMA), and supporting measurements from radiosondes, to quantify aerosol hygroscopic growth. They then seek to determine both what chemical composition promotes hygroscopic growth, and how hygroscopic growth affects visibility. They choose two time periods to contrast their analysis: Case I in the

evenings of 05/21 and Case II on 05/23. These windows were chosen as they had similar water vapor content in the (what I assume to be the boundary layer) air, yet the evening of 05/23 yielded a higher particulate matter (PM) load (∼95 ug/m3 for PM2.5 and ∼70 ug/m3 for PM1) than 05/21 (∼45 ug/m3 for PM2.5 and ∼35 ug/m3 for PM1). From the timeseries in Fig. 3, it appears that Case I is more representative of a normal day in Xingtai whereas Case II is a day of more severe haze. The authors conclude the haze is due to hygroscopic growth (e.g., Fig. 4 e, f, k, l., Fig. 9, and results from the Kasten model).

Overall, the tone of this manuscript is overstated. The work presented is very important and I think should be published. It brings attention to pollution in China's rapidly industrialized cities and includes important techniques that can be deployed elsewhere in China, or eastern/southeast Asia overall. The campaign design is very good! The manuscript is however unfit for publication in Atmos. Chem. Phys. in present form. The authors need more description to show their command over the instrumentation, restructure their sentences to convey ideas more effectively, include error values in their data to support their conclusions, and survey the literature in more depth.

I recommend publication in Atmos. Chem. Phys. after they address several major and minor revisions, suggestions for which are raised below for the authors' consideration.

Major Comments:

Title: Consider an Oxford comma.

Abstract: It appears that the message of the Abstract is that PM caused by anthropogenic activity is more hygroscopic than natural PM, but this does not come across clearly. The English and sentence structure of the entire Abstract require serious revision.

Introduction: The introduction should provide a comprehensive overview of why the manuscript is relevant to the community. The central portion of the introduction should

be moved to experimental techniques, and more literature survey or relevance should be included in the body of the introduction. For example, what other events lead to haze? Is there any proportion that suggests hygroscopic growth is a minor, moderate, or major contributor to haze – either in general or specifically in the Hebei province? What consequences does this intense haze have, either in Hebei, or elsewhere in the world?

Field campaign and instruments: I would merge this section with the Methodology section. The authors don't explain why the time resolution of the lidar is 15 min, and this may be of interest to the reader and would show that the authors have tailored their use for their study. Do other lidar users (whether they use it form the ground or satellite) use 15 min time resolution? Does it depend on how clean the air is? Is there a relationship between altitude and signal-to-noise, particularly at high altitude bins? Later we discover the AE and depolarization ratio, but they haven't been mentioned in this section. This needs to be explained in greater detail to be suitable for publication. Why are the three wavelengths chosen? What does "atmospheric Mie scattering signals" mean? That phrasing is far too vague. Are the authors hinting that UV telescopes detect Mie scattering, perhaps expected from small water droplets or aqueous aerosols? Based on wavelengths used and expected aerosol size ranges, is scattering supposedly in the Mie regime? These questions should be outlined clearly for the interested reader. Finally, whilst detailed information on the ACSM can be found elsewhere, a brief overview is necessary. Ionization scheme? Quad or TOF detection? Can an ACSM measure refractory chloride? Is this an issue? Etc. Furthermore, why is there no mention to the TDMA yet? P7 Lines 15-16 aren't enough to justify leaving out basic information of the TDMA, especially if critical data has been obtained from it.

Methodology: As mentioned above, I'd merge this section with the previous one. Whilst the authors have derived equations rather clearly, and the flow of logic is very good in this section, one major comment for this section is to expand on the Fernald method, and to discuss why (7) and (8) are used. What are the advantages and disadvantages

of one versus the other? Which one is more commonly used? Is one more accurate for specific conditions or wavelengths than the other? Also, in Section 3.3, no mention as to how NH4+ mass concentrations are obtained, nor any of the other supporting measurements. Why don't the authors rearrange information e.g. in Section 3.4. to here, but more importantly, why do the authors leave out so much information on the ACSM? Also, I got lost in clearly understanding what Case I and Case II are supposed to represent. The authors need to rearrange the writing so that one sentence can describe clearly the difference between the two. As I understand, Case I is a clean day and Case II is a hazy day, and that information is clear in the Abstract but not in the body of the document. The authors also need to draw clearer attention to the fact that Case I and II were chosen on days of similar water vapor content, which is hard to understand from the text.

Results and discussion: The major concern here is that no inferences are made, except for the wind directions and the possible airmass sources for the two cases. Describing the results is insufficient for this section.

-First paragraph of page 14: the authors describe their results and conclude 'good mixing atmospheric conditions' for both cases. They fail to explain why that's important for the measurements though. Please explain why good mixing is necessary, or what does it tell us. Parcel is stable? Important for data retrieval? Is there any mention to boundary layer height? How does dilution affect aerosol load / visibility? Why are heights for Case I and II different?

-First paragraph on page 15: overstated. Also, if the authors conclude the haze is likely (from lidar data e.g., Fig. 4 e, f, k, and l) due to hygroscopic growth, I would bolster my minor comment for Fig. 4 that the data visualization is weak to support this claim / conclusion.

-Figure 7: there is no decryption on how NH4+ has been predicated.

-P16 Line 21 – P17 Lines 1-2: I question the validity of this assertion. If I understand

correctly, the authors conclude the aerosols in Case II are more acidic than Case I based on the regression slopes in Fig. 7. This is overstated, particularly if it comes only from one very short time window throughout the campaign. I would challenge it's 'consistent with the results presented'. What are the errors in the slope?

I think the results can support the conclusion, but data visualization needs to be clearer and include some error analysis of sorts, whether it be a confidence interval or standard deviation. Finally, equations for kappa evaluation need to be clearly stated with the proper values from ACSM, or clearly mentioned in the text.

Conclusion: No major comments that aren't addressed in the previous section.

Minor Comments:

P2 Line 1: Remove "particles" and make "aerosol" plural.

P2 Lines 3-4: Rephrase "…the hygroscopic growth effect…" to "…hygroscopic growth…".

P2 Line 4: What do the authors mean by "contributing factors"?

P2 Lines 4-5: "rich measurements" is poor phrasing in English. The sentence in general should be refined.

P2 Line 5: Include "a city" after the comma.

P2 Line 6: This may refer to multiple instances throughout the document, but I encourage the authors to double check any improper usage of words. "most serious" Is a very poor word choice, please revise. A possible solution includes "…suffers from persistent haze…", but words to that effect are encouraged.

P2 Line 7: To stay in line with the tense of the Abstract, perhaps change "are" to "were". Also, was the lidar ground-based or satellite-based? If the lidar was ground-based, please rearrange the sentence or remove "ground-based instruments".

P2 Lines 8-9: Perhaps add 'PM' as the acronym of particulate matter? Also, what type of diameter, presumably aerodynamic diameter?

P2 Line 11: I am not sure what the authors mean by "the evolution". Are they referring to aerosol growth?

P2 Lines 13-16: I'd describe Case I before Case II and remove the unnecessary colon and replace it with 'of', for example.

P2 Line 16: Maybe use the plural, 'were'?

P2 Line 17: Why is 'cases' not capitalized?

P2 Line 18: What is an aerosol acidity value?

P2 Line 22: Why keep both 'aerosol' and 'particle'?

P3 Line 2: I'd replace "Under the same water vapor conditions..." with "For similar ambient RH..."

P3 Lines 2-5: Please rephrase or merge the two sentences using simpler language.

P4 Line 2: Again, no need for "aerosol particles", please rephrase to "Atmospheric aerosols"

P4 Lines 2-4: Reconsider the citation – as well as sentence structure – since what the authors state is well-accepted. If a citation is necessary, one of the classical textbooks should do. Also, please avoid phrasing like "most important". The sentence could be rephrased, e.g.: "Atmospheric aerosols help regulate Earth's climate, mainly by scattering or absorbing incoming radiation" or words to that effect.

P4 Lines 4-6: Again, citation may not be necessary, and both arguments can be collapsed in one sentence and citing a classical aerosol / optics textbook.

P4 Lines 8-11: Please review sentence structure, because I'm assuming what the authors wrote is not what the authors mean. Hygroscopic growth is self-explanatory, the

result however is that the scattering properties change (I would challenge the authors that it strictly increases scattering efficiency or scattering overall).

P4 Line 13: Remove "crucial" . . . as for that matter, the second half of the sentence is a repetition.

Lines 15-16: Please revisit the sentence structure, and rephrase e.g., ". . .water uptake. This growth can be detected by. . ."

P4 Line 15 – P5 Line 18: Please see my major comments. This should belong to the experimental section.

P5 Line 20: Please rephrase ". . .gain deeper insights. . .".

P5 Line 22: Please rephrase "A specific goal. . .".

P6 Lines 3-4: Rephrase "together with other suites of instruments measuring a variety of aerosol properties." with ". . .coupled with supporting measurements."

P6 Lines 12-14: Restructure the sentence by beginning with "Raman lidar was used. . .".

P6 Line 15: This is important and may apply to more instances throughout the document. What does "aerosol optical property" mean? It is far too vague when describing data retrieval. Do the authors mean scattering efficiency? Scattering cross-section? Scattering intensity? Extinction coefficients? Please avoid these generalizations; they are not suited for publication and imply poor working knowledge by the authors.

P6, second paragraph: This needs to be revisited. Please see my major comments regarding this section.

P7 Line 1: Change "Collocated" to "Co-located". Please check throughout.

P7 Line 4: Perhaps mention that the ascension velocity was 'typically' 5-6 m/s?

P7 Line 5: Again, please change "collocated" to "co-located".

P7 Line 17: BJT has already been defined so I'd encourage the authors to be consistent with their acronym usage and replace "Beijing local time" with this acronym.

P8 Line 10: What does "...signal is affected by radiation..." mean?

P9 Line 5: Please insert "reasonably" such that the phrase reads "...agree reasonably well..."

P9 Line 8: If the authors state a percent error for relative humidity, why do they state an absolute error for W? Also, in ref. to equations (3) – (5), can the authors be clear why they choose to display two separate figures for W and RH? Why not combine in one figure, or why report two separate ones at all? Is it for lidar retrieval validation? Unless the percent error for RH is the actual units, not a relative (percentage-weighted) error?

P9 Line 11: Delete the first sentence, it's redundant in my opinion.

P9 Line 16: Could the authors add a little more information regarding the Fernald method for the readers?

P10 Line 8: What is a "hygroscopic parameter"? Are they empirical fits? Do they have a physical meaning? The authors cite some literature, yet do not mention quite exactly what a and b are.

P10 Line 15: Again, vague wording like "...is a key parameter..." should be avoided. Furthermore, is the literature scarce on aerosol acidity and hygroscopic growth? I'd encourage the authors to find more relevant literature to cite.

P10 Line 16: "...aerosols in the atmosphere tend to be more hygroscopic than their neutralized form..." is awfully similar phrasing to the cited literature, Zhang et al., Environ. Sci. Technol., 2007. Apart from the awfully similar phrasing, isn't there a better reference? Might I suggest, but not limit to, Zhang et al., Atmos. Chem. Phys, 2015 (doi: 10.5194/acp-15-8439-2015)?

P10 Lines 17-18: Why does high hygroscopicity of aerosols enhance light scattering?

Also, remove "particles" and make "aerosol" plural (check throughout). Finally, relate this to P4 Lines 8-11, are the authors being consistent?

P11 Line 21: Is chloride not considered because its concentration is extremely low or because the ACSM only measures non-refractory chloride? Is this even an issue for a city as inland as Xingtai?

P12 Line 15: Perhaps "e.g., Tobin et al., 2012"?

P12 Line 16: I would replace "temporal evolution" with "timeseries".

P12 Lines 17-18: Is this a qualitative inference from the authors, or can they provide a correlation of sorts to support their claim?

P13 Line 3: Please rephrase "...and since...".

P12 Lines 3-4: Citing one source hardly makes hygroscopicity 'highly' dependent on the composition of the aerosol. Please rephrase or support with data.

P12 Lines 5-6: Unclear. Are the authors implying that hygroscopic growth diluted the organic fraction (on a mass basis) detected by the ACSM?

P13 Line 17: "...cropped up." is not appropriate. Please change.

P13 Line 18: Remove "a", and I would challenge Case I and Case II help 'fully' understand the phenomenon.

P14 Line 4: Why are these altitude ranges chosen? If the point of the article is to assess haze as a health or visibility issue, wouldn't it make sense to take data below the boundary layer? Or are these heights below the boundary layer?

P13 Line 21 – P14 Line 1: Please rephrase, awkward sentencing.

P14 Lines 2-3: Can the authors explain either now or in the previous section why the AE and depolarization ratio are useful? What do they tell us?

P14 Line 15: Fix "collocated" to "co-located", as per previous comments, unless the

doppler lidar is collocated somewhere? Also, no need to specify again the range of the 'region of interest'.

P14 Lines 13-20: Lots of repetition, unclear and inconsistent sentence structuring, and improper use of citation, and if the authors wish to keep the citation, a more appropriate description or analysis of said 'source region' is required.

P14 Line 21: Perhaps the authors can use a symbol (abbreviation) for the scattering coefficients, rather than the words?

P15 Line 12: Remove "Specifically".

P15 Line 20: Remove "distinctly".

P16 Lines 6-7: Is this an accurate representation for aerosols in Xingtai, Hebei, or Northern China overall, or it's just a generalization? I'm not sure in the current state Lines 6-9 are necessary in this manuscript.

P16 Lines 12-13: Remove sentence.

P17 Line 1: Again, please correct "aerosol particles".

P17 Lines 19-21: Again, this is overstated. How different are kappa value of 0.557 vs. 0.610? I would encourage the authors to use phrasing like 'suggests' or 'point towards', rather than definitive conclusions, which I don't think can be made from the presented data.

P17 Line 22: Remove "ion", fix "aerosol particles", and remove "significant".

P18 Line 3: Replace "Concerning the aerosol scattering enhancement factor, during the last decade, many..." with "In the last decade, many..."

P18 Line 4: Is the nephelometer an example? Or have all studied used the nephelometer?

P18 Line 6: Replace "for use in" with "for".

P18 Line 7: Please fix "aerosol particles".

P18 Line 12: Please fix "aerosol particles".

P18 Lines 13-14: Is the 'kappa model' supposed to be capitalized?

Tables and Figures:

Figure 1: The radiosonde line does not look dashed to me, neither does it look dashed in the legend. Please amend how the authors see fit.

Figure 2: Please keep consistency with data display. Traces should appear like they do in Fig. 1. I don't know if displaying the difference is useful, unless at those heights where the difference is marked, it implies poor lidar performance? If so, please reflect in the main text, because it is arguable how well they agree (as per main text, P9 Lines 4-5)

Figure 3: In the caption, please explain the missing data.

Figure 4: To be consistent with the text, change "Angstrom" to "Ångström", and I don't know if the heights for Cases I and II should be reported to 1 decimal place, unless that is instrument precision (appears so e.g., from P14 Line 4)? Finally, I don't know how impactful this figure is visually if the x-axes are different for Case I and II. I would suggest either keeping x-axes consistent or overlapping the traces for the two cases in one plot. Differences aren't obvious in the current display.

Figure 5: No major comments.

Figure 6: No major comments.

Figure 7: No major comments.

Figure 8: No major comments.

Figure 9: Please fix "Aerosol particle", but more importantly, in the caption, explain this is not data, but a model based on Eq. 3 (as per the text, unless I'm mistaken).

Tables 1-3: No comments.

Table 4: The results weakly support the conclusions of the document. I would encourage the authors to be more transparent with their data, perhaps in a Supporting Information section. Any simple errors to report, e.g. 95% confidence intervals? How was the raw data from the H-TDMA obtained? A timeseries to serve as example perhaps?

References:
* * *

---

## Referee Comment (RC2) · Anonymous Referee #2 · 23 Sep 2018

The manuscript studied aerosol hygroscopic growth through sets of measurements made in Xingtai, Hebei province of China, which suffers from very serious pollutions. Using different instruments, including Raman lidar, handheld particle/mass meter for PM1, PM2.5, aerosol chemical speciation monitor (ACSM), and hygroscopic tandem differential mobility analyzer (H-TDMA), the authors obtained four different quantities representing aerosol hygroscopicity from various aspects. They analyzed the aerosol backscattering enhancement factor [f(RH)] (derived from Raman lidar measurement), the aerosol particle growth factor (derived from the H-TDMA), the aerosol acidity and the hygroscopicity parameter (derived from chemical speciation), of a relatively clean case and a pollution case under similar atmospheric relative humidity, and concluded

that aerosol hygroscopic growth was one of the major factors contributing to heavy haze pollution. The experiment of simultaneous measurements from the four instruments is well designed and the result is important and worth publishing. However I think the manuscript can be improved in writing (English, connection between parts and reasoning), and results can be made more useful if a little more analysis can be done. I recommend publication after addressing the following comments:

1. Add more descriptions or details on the instruments, e.g, what are measured directly and what are derived, uncertainties in their measured/derived quantities.

2. The four variables representing aerosol hygroscopicity from different aspects, namely the aerosol backscattering enhancement factor [f(RH)], the aerosol particle growth factor, the aerosol acidity and the hygroscopicity parameter. To what extent are these variables correlated? What are the correlations among f(RH) , and Äÿ and acidity? Under what circumstances? The answers would make the manuscript potentially more useful, e.g. for aerosol modeling. If possible, expand studied RH range, as I understand there is available data (below the selected loft layers and with lower RH) from the measurements.

3. Figure 3 is an important figure for this manuscript, however I find it is hard to read or draw conclusions with it. And the description of Fig 3. is lack of clarity. Why and how the two cases are selected are poorly demonstrated in the text. Consider adding time series of surface water vapor mixing ratio and RH, as PM and chemical composition data are both obtained at the surface, and the authors are trying to draw some relationships between surface RH and these aerosol data.

4. Page 9. Line 4-8, The authors give absolute errors of Raman-lidar-derived relative humidity and water vapor mixing ratio for a relatively dry case (20%<RH<35% ) in Fig 1 and 2. However boundary layers are generally wetter, and the two case selected for the study both occurred under atmospheric environments with RH>80%. What is the error of Raman-lidar derived RH for wet environment? It would be more meaningful to

add a relatively wet case for validation. Also what are other uncertainties from Raman Lidar, e.g, AE, depolarization ratio? Is there any difference in uncertainty lower and higher altitudes?

Other comments:

Page 7. Line 10-11, Please specify if the handheld particle/mass meter (PC-3016A) measures dry mass or total mass (including water uptake).

Page 11. Line 12-13, "When AV=1.25, 50% of the total sulfate icons in the atmosphere consists of ….and 50% consist of …" This sounds definite. Isn't this just a possible combination of different chemical components?

Page 11 line 20, There are two "because" in this sentence, making it awkward.

Page 13, line 3-5 the whole sentence, starting with "To see if this is the true", reads awkward.

Page 13, line 5-7, "As W in the lower atmospheric layer and the mass concentrations of PM1 and PM2.5 increased, the proportion of organic aerosols decreased, suggesting that the proportion of hygroscopic aerosols increased." This relationship is not straight-forward by looking at Fig 3. Please think of a way of pointing to the readers where to look, maybe by marking these cases. Also in the next paragraph, two cases are selected. Consider adding two vertical lines across Fig a-b-c) so that data can be better visualized.

Page 13, line 8-18, This paragraph is related to Figure 3 and is lack of clarity. "this re-lationship" in the first sentence needs to be explicitly defined. Line 9 "relatively higher", what does it compare to? The first sentence implies the two cases are similar because "this relationship was not seen" in the two instances. However reading along, there seems to be differences for the two cases. What are the similarities and what are the differences for the two cases are not clearly stated in this paragraph. Why are they selected as the studied cases?

Page 14, line 4. How and why are these altitude ranges are selected? I understand that the authors choose RH=80% as the reference RH. Why don't use the whole well-mixed boundary layer, which can give a wider range of RH? I would be curious to see the hygroscopic growths under a wider range of RH. Would the regression relationships between RH and the various hygroscopic growth factors still be valid? If not, how much deviation there would be? Just thinking from aerosol modeling point of view, the result of this paper is potentially applicable in model parameterizations of aerosol hygroscopic growth if a wider RH range can be studied.

Figure 7. If I understand correctly, the reverse of the slopes of fitted lines would be the Acid Value (AV). So maybe consider switch x and y axes for this figure. Then the slopes would be the AV with no need to calculate the reverses.

Page 14, 19-20, "This suggests that aerosol particles were transported to Xingtai from the same source region". Are there local emissions? Can it be excluded?

Page 18. Line 5 "a positive result". Please be explicit.

Page 20, line 2. I think the authors meant relative humidity by "water vapor content". This may have appeared in other places in the draft. Please don't mix use.

---

## Author Comment (AC1) · 14 Nov 2018

**RESPONSES TO REVIEWER #1 COMMENTS**

**General Response:**

We appreciate the reviewer's comments on the manuscript entitled "Aerosol hygroscopic growth, contributing factors, and impact on haze events in a severely polluted region in northern China". All comments are highly valuable and helpful for us to improve our manuscript. We have studied them carefully and have addressed them in the revised manuscript which includes additional investigations. Below are point-by-point responses to the reviewer's comments.

Major Comments:

Title: Consider an Oxford comma.

Abstract: It appears that the message of the Abstract is that PM caused by anthropogenic activity is more hygroscopic than natural PM, but this does not come across clearly. The English and sentence structure of the entire Abstract require serious revision.

**Response**: We have rewritten the abstract so that the findings are more clearly stated. The English has also been improved.

Introduction: The introduction should provide a comprehensive overview of why the manuscript is relevant to the community. The central portion of the introduction should be moved to experimental techniques, and more literature survey or relevance should be included in the body of the introduction. For example, what other events lead to haze? Is there any proportion that suggests hygroscopic growth is a minor, moderate, or major contributor to haze – either in general or specifically in the Hebei province? What consequences does this intense haze have, either in Hebei, or elsewhere in the world?

**Response:** We have extensively revised the introduction by adding a comprehensive overview of why the study is relevant to the community. We have also done a more extensive literature survey about haze events and have added more descriptions about experimental techniques. The latter provides a brief overview of field instruments used to study aerosol hygroscopic growth. Aerosol hygroscopic growth has a major impact on haze events, but there are many other factors leading to haze such as emissions, weather conditions, planetary boundary layer (PBL)-aerosol interactions, and aerosol chemical and physical properties.

**A comprehensive overview of why the study is relevant to the community:**

"Aerosols, as solid or liquid particles suspended in the air, help regulate Earth's climate mainly by directly scattering or absorbing incoming radiation, or indirectly changing cloud optical and microphysical properties (IPCC, 2013). Many studies suggest that aerosols have a direct impact on human health (Araujo et al., 2008;

Anenberg et al., 2010; Liao et al., 2015; Li et al., 2017). For example, exposure to fine airborne particulates is linked to increased respiratory and cardiovascular diseases (Hu et al., 2015). Atmospheric aerosols can also reduce visibility. Poor visibility is not only detrimental to human health but also hazardous to all means of transportation (Zhang et al., 2010; Zhang et al., 2018)." Therefore, studies on aerosol formation and its influence are important for predicting climate change and improving the human habitat.

**Other factors influencing haze:**

"Poor visibility is caused by the presence of atmospheric aerosols whose loading depends on both emission and meteorology. The increase in anthropogenic emissions directly affects the formation of haze, such as biomass burning, and factory and vehicle emissions (Watson, 2002; Sun et al., 2006; Q. Liu et al., 2016; Qu et al., 2018). During some major events like the 2008 Summer Olympic Games, drastic measures were taken to reduce emissions which led to a significant improvement in air quality (Huang et al., 2014; Shi et al., 2016; Y.-Y. Wang et al., 2017). This attests to the major role of emissions in air quality. Surface solar radiation and weather such as wind conditions also affect aerosol pollution (Yang et al., 2015). It has been widely known that aerosols interact with the planetary boundary layer (PBL; Quan et al., 2013; Li et al., 2017; Qu et al., 2018; Su et al., 2018). More aerosols reduce surface solar radiation, resulting in a more stable PBL which enhances the accumulation of pollutants within the PBL. Numerous studies have highlighted that the diurnal evolution of the PBL is crucial to the formation of air pollution episodes (Tie et al., 2015; Amil et al., 2016; Kusumaningtyas and Aldrian, 2016; Li et al., 2017; Qu et al., 2018). Besides feedbacks, the stability of the PBL affects the dispersion of pollutants."

**Roles of the hygroscopic effect:**

"Aerosol hygroscopicity also significantly affects visibility due to the swelling of aerosols (Jeong et al., 2007; Wang et al., 2014). A number of studies have shown that aerosol hygroscopic growth can accelerate the formation and evolution of haze pollution in the North China Plain (NCP; e.g., Quan et al., 2011; Liu et al., 2013; Wang et al., 2014; Yang et al., 2015)."

**More information about experimental techniques to measure the hygroscopicity:**

"There are many ways to measure aerosol hygroscopicity. A widely used parameter, the aerosol particle size hygroscopic growth factor (GF), is defined as the ratio of the wet particle diameter ($D_{p,wet}$) at a high relative humidity (RH) to the corresponding dry diameter ($D_{p,dry}$). The GF at a certain particle size can be detected by a hygroscopicity tandem differential mobility analyzer (H-TDMA; e.g., Liu et al., 1978; Swietlicki et al., 2008; Y.-Y. Wang et al., 2017). In general, the H-TDMA system mainly consists of two differential mobility analyzer (DMA) systems and one condensation particle counter (CPC). The DMA is first used to select particles at a specific size, and the second DMA and the CPC are used to measure the size distribution of humidified particles. Another instrument known as the differential aerosol sizing and hygroscopicity spectrometer

probe (DASH-SP) can also measure the GF at different RHs (Sorooshian et al., 2008). The DASH-SP couples one DMA and an optical particle size spectrometer (OPSS). The dry size-dependent particles are selected by the DMA, then exposed to different RH environments and finally measured in the OPSS (Sorooshian et al., 2008; Rosati et al., 2015)."

Field campaign and instruments: I would merge this section with the Methodology section. The authors don't explain why the time resolution of the lidar is 15 min, and this may be of interest to the reader and would show that the authors have tailored their use for their study. Do other lidar users (whether they use it form the ground or satellite) use 15 min time resolution? Does it depend on how clean the air is? Is there a relationship between altitude and signal-to-noise, particularly at high altitude bins? Later we discover the AE and depolarization ratio, but they haven't been mentioned in this section. This needs to be explained in greater detail to be suitable for publication. Why are the three wavelengths chosen? What does "atmospheric Mie scattering signals" mean? That phrasing is far too vague. Are the authors hinting that UV telescopes detect Mie scattering, perhaps expected from small water droplets or aqueous aerosols? Based on wavelengths used and expected aerosol size ranges, is scattering supposedly in the Mie regime? These questions should be outlined clearly for the interested reader. Finally, whilst detailed information on the ACSM can be found elsewhere, a brief overview is necessary. Ionization scheme? Quad or TOF detection? Can an ACSM measure refractory chloride? Is this an issue? Etc. Furthermore, why is there no mention to the TDMA yet? P7 Lines 15-16 aren't enough to justify leaving out basic information of the TDMA, especially if critical data has been obtained from it.

**Response:** We have merged this section with the methodology section in the revised manuscript. In this field campaign, the time resolution was set to 15 min based on original factory settings. These settings have been used in previous studies. A different interval can be used but this would affect the lidar power. The signal-to-noise ratio depends on height. For a ground-based lidar, this ratio is low due to the attenuation of lidar signals. Three wavelengths were chosen to increase our detection ability, noting that there is little choice in the selection of wavelengths in atmospheric lidar applications. This fundamental knowledge was not included in the manuscript to keep the text concise.

We also have added the following new descriptions about the AE and the depolarization ratio.

[revised manuscript text omitted]

Methodology: As mentioned above, I'd merge this section with the previous one. Whilst the authors have derived equations rather clearly, and the flow of logic is very good in this section, one major comment for this section is to expand on the Fernald method, and to discuss why (7) and (8) are used. What are the advantages and disadvantages of one versus the other? Which one is more commonly used? Is one more accurate for specific conditions or wavelengths than the other? Also, in Section 3.3, no mention as to how NH4+ mass concentrations are obtained, nor any of the other supporting measurements. Why don't the authors rearrange information e.g. in Section 3.4. to here, but more importantly, why do the authors leave out so much information on the ACSM? Also, I got lost in clearly understanding what Case I and Case II are supposed to represent. The authors need to rearrange the writing so that one sentence can describe clearly the difference between the two. As I understand, Case I is a clean day and Case II is a hazy day, and that information is clear in the Abstract but not in the body of the document. The authors also need to draw clearer attention to the fact that Case I and II were chosen on days of similar water vapor content, which is hard to understand from the text.

**Response:** We have merged this section with the previous one in the revised manuscript. More information about the Fernald method and equations (7) and (8) were added. The dual-parameter fit equation is similar to the single-parameter equation, but with an additional parameter, i.e., a scale factor. In this study, two commonly used parameterized equations were used to verify the consistency of the results. Results from the model that best fit measurement data are shown in the figures. Only the 532-nm wavelength was considered. This is why the equations are not wavelength-dependent. The mass concentration of the measured $NH_4^+$ was measured by the ACSM. This has been clearly stated in the revised manuscript. We have added more information about the ACSM in the section about instruments. The selection criteria of the cases were rephrased in the revised manuscript, which includes a clear description of the main difference (different pollution conditions) and similarity (same ambient relative humidity) between the two cases.

**Description of the Fernald method:**

"Here, we use the Fernald method to retrieve AECs (Fernald et al., 1972; Fernald, 1984), which is an analytic solution to the following basic lidar equation for Mie scattering:

$$P_s(z) = ECZ^2[\beta_1(z) + \beta_2(z)]T_1^2(z)T_2^2(z), \tag{3}$$

where $P_s(z)$ is the return signal, $E$ is the energy emitted by the laser, $C$ is the calibration constant of the lidar system, and $\beta_1(z)$ and $\beta_2(z)$ are the backscattering cross-sections of atmospheric aerosols and molecules at altitude $z$, respectively. $T_1(z)$ and $T_2(z)$ are the transmittances of aerosols and air molecules at height $z$."

**More description about (7) and (8):**

"Finally, a relationship between $f(RH)$ and RH was established. The most commonly used equations are the single-parameter fit equation (e.g., Hänel, 1980; Kotchenruther and Hobbs, 1998; Gassó et al., 2000) and the dual-parameter fit equation (e.g., Hänel, 1980; Carrico, 2003; Zieger et al., 2011). The single-parameter fit equation introduced by Hänel (1976) is

$$f(RH) = \left(\frac{1-RH}{1-RH_{ref}}\right)^{-\gamma}, \tag{8}$$

where $\gamma$ in an empirical parameter. Larger $\gamma$ values in this formulation denote a stronger hygroscopic growth.

The dual-parameter fit equation is (Fernández et al., 2015)

$$f(RH) = a(1-RH)^{-b}. \tag{9}$$

The single- and dual-parameter fit equations are similar, but with an additional scale factor parameter, $a$, in the case of the dual-parameter fit equation. The parameter $b$ is also an empirical parameter with larger values of $b$ indicating particles with stronger hygroscopicities. In this study, both parameterized equations are used to verify the consistency of the results. The equation that fits the measurement data best is selected."

**Selection of aerosol hygroscopic cases:**

"How aerosol hygroscopic growth cases were chosen is described here. First, atmospheric mixing conditions were examined using radiosonde-based vertical potential temperature ( ) and $W$ profiles. Cases with near-constant values of and $W$ in the analyzed layer (variations less than 2°C and 2 g kg$^{-1}$, respectively) represent good atmospheric mixing conditions (Granados-Muñoz et al., 2015). Then aerosol backscattering coefficient profiles at 532 nm were calculated using the Fernald method (see details in section 2.2.1). A simultaneous increase in atmospheric RH and the aerosol backscattering coefficient is also needed, which might indicate aerosol hygroscopic growth (Bedoya-Velásquez et al., 2018). Based on the above criteria, individual cases with the same ambient humidity and different pollution conditions were selected for studying the influence of aerosol hygroscopicity on haze events."

Results and discussion: The major concern here is that no inferences are made, except for the wind directions and the possible airmass sources for the two cases. Describing the results is insufficient for this section.

**Response:** More inferences have been made in the revised manuscript. More discussion of the results has been added.

-First paragraph of page 14: the authors describe their results and conclude 'good mixing atmospheric conditions' for both cases. They fail to explain why that's important for the measurements though. Please explain why good mixing is necessary, or what does it tell us. Parcel is stable? Important for data retrieval? Is there any mention to boundary layer height? How does dilution affect aerosol load / visibility? Why are heights for Case I and II different?

**Response:** $W$ and $\theta$ variations are less than 2 g kg$^{-1}$ and 2$^{o}$C, respectively, showing that good mixing atmospheric conditions were present in both cases (Granados-Muñoz et al., 2014). This information suggests atmospheric vertical homogeneity in the layers considered in the study. Moreover, we can infer that the increase in the aerosol backscattering coefficient is caused primarily by the increase in RH in the range of values of interest (Veselovskii et al., 2009; Fernández et al., 2015; Granados-Muñoz et al., 2015; Lv et al., 2017). This is why good mixing is necessary.

Why are the heights for Cases I and II different? A simultaneous increase in aerosol backscattering coefficient and RH values is the precondition for determining where the layer is located. In reference to previous studies (e.g., Fernández et al., 2015; Granados-Muñoz et al., 2015; Lv et al., 2017; Bedoya-Velásquez et al., 2018), the boundary layer height was not taken into account in the case selection. Your comments are sound, but the experimental data used in this study are limited and preclude doing what you suggest. We will consider your comment in future work.

-First paragraph on page 15: overstated. Also, if the authors conclude the haze is likely (from lidar data e.g., Fig. 4 e, f, k, and l) due to hygroscopic growth, I would bolster my minor comment for Fig. 4 that the data visualization is weak to support this claim / conclusion.

**Response:** As done in previous studies (e.g., Granados-Muñoz et al., 2015; Lv et al., 2017; Bedoya-Velásquez et al., 2018), what we try to show in this paragraph is that the changes in AE and depolarization ratio gives us more confidence to believe that the increase in aerosol backscattering coefficient with height in each case is primarily due to hygroscopic growth. More details are in the original manuscript on page 14 (lines 1–20).

-Figure 7: there is no decryption on how NH4+ has been predicated.

**Response:** The mass concentration of NH4+ is predicted using Eq. (10) in the revised manuscript.

"Figure 7 shows mass concentrations of measured ammonium (NH$_4$) as a function of predicted NH$_4$ assuming full neutralization of sulfate, nitrate, and chloride during the full day of 21 May 2016 (blue dots, Case I) and 23 May 2016 (green dots, Case II). The solid blue and green lines are the least-squares regression lines for Case I and Case II, respectively. The 1:1 line is shown in red."

-P16 Line 21 – P17 Lines 1-2: I question the validity of this assertion. If I understand correctly, the authors conclude the aerosols in Case II are more acidic than Case I based on the regression slopes in Fig. 7. This is overstated, particularly if it comes only from

one very short time window throughout the campaign. I would challenge it's 'consistent with the results presented'. What are the errors in the slope?

**Response:** We conclude that Case II aerosols are more acidic than Case I aerosols based on the regression slopes and the parameter $AV$ for each case. The data used for the linear fitting of the whole day were obtained by the ACSM, and $AV$ was calculated at the closest time of each case. The root-mean-squared errors of the liner regression best-fit lines are 0.63 and 0.48 on 21 May 2016 and 23 May 2016, respectively. The following has been added to the revised manuscript:

"The acidity of aerosols in Case II is greater than that in Case I, suggesting that aerosols in Case II were more hygroscopic than those in Case I. This is consistent with the results presented in section 3.2.1."

I think the results can support the conclusion, but data visualization needs to be clearer and include some error analysis of sorts, whether it be a confidence interval or standard deviation. Finally, equations for kappa evaluation need to be clearly stated with the proper values from ACSM, or clearly mentioned in the text.

**Response:** In the methodology section, we analyzed the absolute error between the water vapor mixing ratios retrieved by the Raman lidar and the radiosonde. The error bars of the relevant parameters are plotted in Fig. 5. We did not provide the uncertainty of the aerosol backscattering enhancement factor because it relies on the uncertainties of many factors including aerosol properties, ambient RH, hygroscopic growth itself, and so on. The H-TDMA and the ACSM cannot provide an error bar for only one point.

Concerning the $\kappa$ evaluation, we first calculated the number of moles of $SO_4^{2-}$, $NO_3^-$, and $NH_4^+$ based on their mass concentrations obtained by the ACSM. The mass concentrations and mole numbers of the main inorganic salts in $PM_1$ [$NH_4NO_3$, $NH_4HSO_4$, and $(NH_4)_2SO_4$] are then evaluated based on an ion-pairing scheme (Eq. 11 in the original manuscript) and their density values (see Table 1). Finally, the hygroscopicity parameter ($\kappa$) was calculated based on the ZSR mixing rule (Eq. 12 in the original manuscript) using moles of all species and their corresponding hygroscopicity parameters (Table 1). This has been clearly stated in the revised manuscript.

Conclusion: No major comments that aren't addressed in the previous section.

Minor Comments:

P2 Line 1: Remove "particles" and make "aerosol" plural.

Done.

P2 Lines 3-4: Rephrase "…the hygroscopic growth effect…" to "…hygroscopic growth…".

Done.

P2 Line 4: What do the authors mean by "contributing factors"?

**Response:** The "contributing factors" are the main factors affecting the hygroscopic

properties of aerosols.

P2 Lines 4-5: "rich measurements" is poor phrasing in English. The sentence in general should be refined.

Done.

"This study investigates the impact of the aerosol hygroscopic growth effect on haze events in Xingtai, a heavily polluted city in the central part of the North China Plain, using a large array of instruments measuring aerosol optical, physical, and chemical properties."

P2 Line 5: Include "a city" after the comma.

Done.

P2 Line 6: This may refer to multiple instances throughout the document, but I encourage the authors to double check any improper usage of words. "most serious" Is a very poor word choice, please revise. A possible solution includes "…suffers from persistent haze…", but words to that effect are encouraged.

Done.

"This study investigates the impact of the aerosol hygroscopic growth effect on haze events in Xingtai, a heavily polluted city in the central part of the North China Plain …"

P2 Line 7: To stay in line with the tense of the Abstract, perhaps change "are" to "were". Also, was the lidar ground-based or satellite-based? If the lidar was ground-based, please rearrange the sentence or remove "ground-based instruments".

**Response:** The lidar was ground-based.

"Key instruments used and measurements made include the Raman lidar for atmospheric water vapor content and aerosol optical profiles, the PC-3016A GrayWolf six-channel handheld particle/mass meter for atmospheric total particulate matter (PM) that have diameters less than 1 μm and 2.5 μm ($PM_1$ and $PM_{2.5}$, respectively), the aerosol chemical speciation monitor (ACSM) for chemical components in $PM_1$, and the hygroscopic tandem differential mobility analyzer (H-TDMA) for aerosol hygroscopicity."

P2 Lines 8-9: Perhaps add 'PM' as the acronym of particulate matter? Also, what type of diameter, presumably aerodynamic diameter?

Done.

"… a GrayWolf six-channel handheld particle/mass meter for atmospheric total particulate matter (PM) that have aerodynamic diameters less than 1 μm and 2.5 μm ($PM_1$ and $PM_{2.5}$, respectively) …"

P2 Line 11: I am not sure what the authors mean by "the evolution". Are they referring to aerosol growth?

**Response:** The term "the evolution" means the changes. It has been modified in the revised manuscript.

"The changes in $PM_1$ and $PM_{2.5}$ agreed well with that of the water vapor content …"

P2 Lines 13-16: I'd describe Case I before Case II and remove the unnecessary colon and replace it with 'of', for example.

Done.

"The lidar-estimated hygroscopic enhancement factor for the aerosol backscattering coefficient during a relatively clean period (Case I) was lower than that during a pollution event (Case II) with similar relative humidity (RH) of 80–91%."

P2 Line 16: Maybe use the plural, 'were'?

Done.

P2 Line 17: Why is 'cases' not capitalized?

**Re:** It has been modified.

P2 Line 18: What is an aerosol acidity value?

**Response:** The aerosol acidity ($AV$) in this study was defined as follows:

$$AV = (2 \cdot SO_4^{2-} / 96 + NO_3^- / 62 + Cl^- / 35.5) / (NH_4^+ / 18), \qquad (11)$$

where $SO_4^{2-}$, $NO_3^-$, $Cl^-$, and $NH_4^+$ represent the mass concentrations (in $\mu g\ m^{-3}$) of the three species measured by the ACSM. The molecular weights of $SO_4^{2-}$, $NO_3^-$, $Cl^-$, and $NH_4^+$ are 96, 62, 35.5, and 18. Aerosols are considered "bulk neutralized" if $AV = 1$ and "strongly acidic" if $AV > 1.25$. Details in the section 2.2.3 in the revised manuscript.

P2 Line 22: Why keep both 'aerosol' and 'particle'?

**Response:** The term has been modified.

P3 Line 2: I'd replace "Under the same water vapor conditions…" with "For similar ambient RH…"

Done.

"For similar ambient RH levels, the high content of nitrate facilitates the hygroscopic growth of aerosols, which may be a major factor contributing to heavy haze episodes in Xingtai."

P3 Lines 2-5: Please rephrase or merge the two sentences using simpler language.

**Response:** Rephrased. Details in the last response.

P4 Line 2: Again, no need for "aerosol particles", please rephrase to "Atmospheric aerosols"

Done.

P4 Lines 2-4: Reconsider the citation – as well as sentence structure – since what the authors state is well-accepted. If a citation is necessary, one of the classical textbooks

should do. Also, please avoid phrasing like "most important". The sentence could be rephrased, e.g.: "Atmospheric aerosols help regulate Earth's climate, mainly by scattering or absorbing incoming radiation" or words to that effect.

Done.

"Aerosols, as solid or liquid particles suspended in the air, help regulate Earth's climate mainly by directly scattering or absorbing incoming radiation, or indirectly changing cloud optical and microphysical properties (IPCC, 2013)."

P4 Lines 4-6: Again, citation may not be necessary, and both arguments can be collapsed in one sentence and citing a classical aerosol / optics textbook.

Done. Details in the last response.

P4 Lines 8-11: Please review sentence structure, because I'm assuming what the authors wrote is not what the authors mean. Hygroscopic growth is self-explanatory, the result however is that the scattering properties change (I would challenge the authors that it strictly increases scattering efficiency or scattering overall).

**Response:** The sentence has been removed.

P4 Line 13: Remove "crucial" … as for that matter, the second half of the sentence is a repetition.

Done.

"A number of studies have shown that aerosol hygroscopic growth can accelerate the formation and evolution of haze pollution in the North China Plain (NCP; e.g., Quan et al., 2011; Liu et al., 2013; Wang et al., 2014; Yang et al., 2015)."

P4 Lines 15-16: Please revisit the sentence structure, and rephrase e.g., "…water uptake. This growth can be detected by…"

**Response:** It has been rephrased.

 "A widely used parameter, the aerosol particle size hygroscopic growth factor (GF), is defined as the ratio of the wet particle diameter ($D_{p,wet}$) at a high relative humidity (RH) to the corresponding dry diameter ($D_{p,dry}$). The GF at a certain particle size can be detected by a hygroscopicity tandem differential mobility analyzer (H-TDMA; e.g., Liu et al., 1978; Swietlicki et al., 2008; Y.-Y. Wang et al., 2017)."

P4 Line 15 – P5 Line 18: Please see my major comments. This should belong to the experimental section.

**Response:** This passage provides a brief overview of field instruments used for studying aerosol hygroscopic growth. We also mention the advantages of using a lidar instead of other instruments to study aerosol hygroscopicity. Among the instruments mentioned, only the H-TDMA and Raman lidar were used in this study. Details can be found in the replies to the major comments.

P5 Line 20: Please rephrase "…gain deeper insights…".

**Response:** This sentence have been removed.

P5 Line 22: Please rephrase "A specific goal…".

Done.

"The goal of this study is to further investigate how aerosol hygroscopic growth affects haze events and what are the controlling factors by combining surface and vertical measurements of aerosol optical, physical, and chemical properties."

P6 Lines 3-4: Rephrase "together with other suites of instruments measuring a variety of aerosol properties." with "…coupled with supporting measurements."

**Response:** This sentence has been modified.

"The goal of this study is to further investigate how aerosol hygroscopic growth affects haze events and what are the controlling factors by combining surface and vertical measurements of aerosol optical, physical, and chemical properties."

P6 Lines 12-14: Restructure the sentence by beginning with "Raman lidar was Used…".

Done.

"A Raman lidar was used to analyze the relationship between atmospheric water vapor content and $PM_1$ or $PM_{2.5}$ mass concentrations, …"

P6 Line 15: This is important and may apply to more instances throughout the document. What does "aerosol optical property" mean? It is far too vague when describing data retrieval. Do the authors mean scattering efficiency? Scattering cross-section? Scattering intensity? Extinction coefficients? Please avoid these generalizations; they are not suited for publication and imply poor working knowledge by the authors.

**Response:** The "aerosol optical property" means the aerosol extinction and backscattering coefficients, Ångström exponent (AE), and the depolarization ratio. This has been stated in the revised manuscript.

P6, second paragraph: This needs to be revisited. Please see my major comments regarding this section.

**Re:** It has been revised according to your comments.

P7 Line 1: Change "Collocated" to "Co-located". Please check throughout.

Done.

P7 Line 4: Perhaps mention that the ascension velocity was 'typically' 5-6 m/s?

Done.

P7 Line 5: Again, please change "collocated" to "co-located".

Done.

P7 Line 17: BJT has already been defined so I'd encourage the authors to be consistent with their acronym usage and replace "Beijing local time" with this acronym.

Done.

P8 Line 10: What does "…signal is affected by radiation…" mean?

**Response:** In the daytime, the strong background light will reduce the original signal-to-noise ratio of the Raman signal.

P9 Line 5: Please insert "reasonably" such that the phrase reads "…agree reasonably Well…"

Done.

"The $W$ profiles agree reasonably well with an absolute error between them less than 0.5 g kg$^{-1}$."

P9 Line 8: If the authors state a percent error for relative humidity, why do they state an absolute error for W? Also, in ref. to equations (3) – (5), can the authors be clear why they choose to display two separate figures for W and RH? Why not combine in one figure, or why report two separate ones at all? Is it for lidar retrieval validation? Unless the percent error for RH is the actual units, not a relative (percentage-weighted) error?

**Response:** The percent error of RH is the actual unit, not a relative (percentage-weighted) error. Figures 1 and 2 have been merged.

P9 Line 11: Delete the first sentence, it's redundant in my opinion.

Done.

P9 Line 16: Could the authors add a little more information regarding the Fernald method for the readers?

**Response:** We have added some information about the Fernald method.

"Here, we use the Fernald method to retrieve AECs (Fernald et al., 1972; Fernald, 1984), which is an analytic solution to the following basic lidar equation for Mie scattering:

$$P_s(z) = ECZ^{-2}[\beta_1(z) + \beta_2(z)]T_1^2(z)T_2^2(z), \qquad (3)$$

where $P_s(z)$ is the return signal, $E$ is the energy emitted by the laser, $C$ is the calibration constant of the lidar system, and $\beta_1(z)$ and $\beta_2(z)$ are the backscattering cross-sections of atmospheric aerosols and molecules at altitude $z$, respectively."

P10 Line 8: What is a "hygroscopic parameter"? Are they empirical fits? Do they have a physical meaning? The authors cite some literature, yet do not mention quite exactly what a and b are.

**Response:** The single- and dual-parameter fit equations are similar, but with an additional scale factor parameter, $a$, in the case of the dual-parameter fit equation. The parameter $b$ is also an empirical parameter with larger values of $b$ indicating particles with stronger hygroscopicities.

P10 Line 15: Again, vague wording like "…is a key parameter…" should be avoided. Furthermore, is the literature scarce on aerosol acidity and hygroscopic growth? I'd encourage the authors to find more relevant literature to cite.

**Response:** The sentence has been rephrased, and more relevant literature has been cited.

"Aerosol acidity is associated with aerosol hygroscopic growth (e.g. Sun et al., 2009; Fu et al., 2015; Zhang et al., 2015; Lv et al., 2017)."

P10 Line 16: "…aerosols in the atmosphere tend to be more hygroscopic than their neutralized form…" is awfully similar phrasing to the cited literature, Zhang et al., Environ. Sci. Technol., 2007. Apart from the awfully similar phrasing, isn't there a better reference? Might I suggest, but not limit to, Zhang et al., Atmos. Chem. Phys, 2015 (doi: 10.5194/acp-15-8439-2015)?

**Response:** This sentence has been rephrased as follows.

"When atmospheric aerosols are acidic, they have stronger hygroscopicities than when in their neutralized forms (Zhang et al., 2015)."

P10 Lines 17-18: Why does high hygroscopicity of aerosols enhance light scattering? Also, remove "particles" and make "aerosol" plural (check throughout). Finally, relate this to P4 Lines 8-11, are the authors being consistent?

**Response:** It has been rephrased as follows.

"The swelling of aerosols due to hygroscopic growth enhances their ability to scatter solar radiation."

P11 Line 21: Is chloride not considered because its concentration is extremely low or because the ACSM only measures non-refractory chloride? Is this even an issue for a city as inland as Xingtai?

**Response:** As mentioned above, the ACSM only measures non-refractory aerosol species including non-refractory chloride. According to other studies using a similar method (e.g., Gysel et al., 2007; Wu et al., 2016; Liu et al., 2016; Y.-Y. Wang et al., 2018), chlorine ions are usually ignored when estimating the hygroscopicity parameter using the ZSR mixing rule (Eq. 12 in the original manuscript) because its concentration is always low. Zhang et al. (2018) have found that the mass concentration of chloride is low in Xingtai based on the $PM_1$ species there.

P12 Line 15: Perhaps "e.g., Tobin et al., 2012"?

Done.

P12 Line 16: I would replace "temporal evolution" with "timeseries".

Done.

"Figures 3b and 3c show the simultaneous time series of the surface mass concentrations of $PM_1$ and $PM_{2.5}$, and $W$ and RH, respectively."

P12 Lines 17-18: Is this a qualitative inference from the authors, or can they provide a correlation of sorts to support their claim?

**Response:** Yes, this is a qualitative inference.

P13 Line 3: Please rephrase "…and since…".

Done.

"And Zou et al. (2018) shows that aerosol hygrocopicity is related to aerosol chemical composition over the North China Plain."

P13 Lines 3-4: Citing one source hardly makes hygroscopicity 'highly' dependent on the composition of the aerosol. Please rephrase or support with data.

Done.

"And Zou et al. (2018) shows that aerosol hygrocopicity is related to aerosol chemical composition over the North China Plain."

P13 Lines 5-6: Unclear. Are the authors implying that hygroscopic growth diluted the organic fraction (on a mass basis) detected by the ACSM?

**Response:** No. Inorganics are the main aerosol components contributing to aerosol hygroscopicity (Liu et al., 2014). The decrease in the organics fraction suggests the increase in the inorganics fraction (hygroscopic aerosols).

P13 Line 17: "…cropped up." is not appropriate. Please change.

Done.

"… but they ignored some unexpected cases behind this positive correlation."

P13 Line 18: Remove "a", and I would challenge Case I and Case II help 'fully' understand the phenomenon.

Done.

"The two unexpected cases that occurred on 21 May 2016 (Case I) and 23 May 2016 (Case II) were selected for further study."

P14 Line 4: Why are these altitude ranges chosen? If the point of the article is to assess haze as a health or visibility issue, wouldn't it make sense to take data below the boundary layer? Or are these heights below the boundary layer?

**Response:** A simultaneous increase in aerosol backscattering coefficient and RH values is the precondition for determining where the layer is located. In reference to previous studies (e.g., Fernández et al., 2015; Granados-Muñoz et al., 2015; Lv et al., 2017; Bedoya-Velásquez et al., 2018), the boundary layer height was not taken into account in the case selection. Your comments are sound, but the experimental data used in this study are limited and preclude doing what you suggest. We will consider your comment in future work.

P13 Line 21 – P14 Line 1: Please rephrase, awkard sentencing.

Done.

"Two cases were selected: one on 21 May 2016 (Case I) and the other on 23 May 2016 (Case II) at the closest time of the radiosonde launch time at 1915 BJT."

P14 Lines 2-3: Can the authors explain either now or in the previous section why the AE and depolarization ratio are useful? What do they tell us?

**Response:** Based on your previous comments, detailed information about the AE and the depolarization ratio has been added to the instruments section. The last paragraph on page 14 of the original manuscript also has a description.

"Based on the perpendicular and parallel components at 532 nm received by the lidar system, the aerosol depolarization ratio, a parameter that measures the shapes of aerosols, can be calculated. In general, the more irregular the particle shape, the larger the value of the depolarization ratio (Chen et al., 2002; Baars et al., 2016). The AE can also be calculated using lidar signals at 532 and 1064 nm, which is inversely related to the average size of the aerosols (Ångström, 1964; Tiwari et al., 2016)."

P14 Line 15: Fix "collocated" to "co-located", as per previous comments, unless the doppler lidar is collocated somewhere? Also, no need to specify again the range of the 'region of interest'.

Done.

P14 Lines 13-20: Lots of repetition, unclear and inconsistent sentence structuring, and improper use of citation, and if the authors wish to keep the citation, a more appropriate description or analysis of said 'source region' is required.

**Response:** The revised paragraph is as follows.

"Figure 4 shows the time series of the horizontal wind velocity and direction retrieved from the co-located Doppler lidar system. From 1830–2030 BJT, Case I (Fig. 4c) and Case II (Fig. 4d) winds within their respective layers are mainly from the north and northwest, respectively, and have relatively low speeds (< 5 m s$^{-1}$, Fig. 4a and 4b). This suggests that the aerosols in each case were transported into their respective layers at low speeds from almost the same direction. In other words, there is no change in the aerosol type of both cases within the region of interest."

P14 Line 21: Perhaps the authors can use a symbol (abbreviation) for the scattering coefficients, rather than the words?

Done.

"The RH and $_{532}$ simultaneously increase with altitude in the Case I (Fig. 5c and 5d) and Case II (Fig. 5i and 5j) layers of interest."

P15 Line 12: Remove "Specifically".

Done.

P15 Line 20: Remove "distinctly".

Done.

P16 Lines 6-7: Is this an accurate representation for aerosols in Xingtai, Hebei, or Northern China overall, or it's just a generalization? I'm not sure in the current state Lines 6-9 are necessary in this manuscript.

**Response:** It is a generalized description. We have removed these sentences in the revised manuscript.

P16 Lines 12-13: Remove sentence.

Done.

P17 Line 1: Again, please correct "aerosol particles".

Done.

P17 Lines 19-21: Again, this is overstated. How different are kappa value of 0.557 vs. 0.610? I would encourage the authors to use phrasing like 'suggests' or 'point towards', rather than definitive conclusions, which I don't think can be made from the presented data.

Done.

"This suggests that the aerosol hygroscopicity for Case II was higher than that for Case I."

P17 Line 22: Remove "ion", fix "aerosol particles", and remove "significant".

Done.

"It also suggests that under the same ambient RH conditions, the nitrate content in aerosols can cause differences in the hygroscopicity of aerosols."

P18 Line 3: Replace "Concerning the aerosol scattering enhancement factor, during the last decade, many…" with "In the last decade, many…"

Done.

"In the last decade, many studies have compared remotely sensed and in situ aerosol scattering enhancement factor measurements using a humidified tandem nephelometer and have shown positive results (Zieger et al., 2011, 2012; Sheridan et al., 2012; Tesche et al., 2014; Lv et al., 2017)."

P18 Line 4: Is the nephelometer an example? Or have all studied used the nephelometer?

**Response:** To our knowledge, almost all studies have used the nephelometer (e.g., Zieger et al., 2011, 2012; Sheridan et al., 2012; Tesche et al., 2014; Lv et al., 2017).

P18 Line 6: Replace "for use in" with "for".

**Done.**

The H-TDMA is also a reliable instrument for measuring the aerosol hygroscopicity due to water uptake (Liu et al., 1978).

P18 Line 7: Please fix "aerosol particles".

Done.

The aerosols diameter GFs observed by the ground-based H-TDMA at the closest time of each case are examined next.

P18 Line 12: Please fix "aerosol particles".

Done.

P18 Lines 13-14: Is the 'kappa model' supposed to be capitalized?

**Response:** We used the symbol instead.

Tables and Figures:

Figure 1: The radiosonde line does not look dashed to me, neither does it look dashed in the legend. Please amend how the authors see fit.

Done.

Figure 2: Please keep consistency with data display. Traces should appear like they do in Fig. 1. I don't know if displaying the difference is useful, unless at those heights where the difference is marked, it implies poor lidar performance? If so, please reflect in the main text, because it is arguable how well they agree (as per main text, P9 Lines 4-5)

**Re:** Figures 1 and 2 were merged. The biases in Fig. 1 and Fig. 2 (original manuscript) are both absolute errors.

[Figure]

Fig. 1. (a, c) Water vapor mixing ratios ($W$) and RH profiles at 0515 BJT 24 May 2016 retrieved by the Raman lidar (blue line) and the radiosonde (red dashed line), respectively, and (b, d) the absolute error in $W$ and RH between the lidar and radiosonde retrievals (lidar minus radiosonde), respectively.

Figure 3: In the caption, please explain the missing data.

Done.

"Blank parts of the data are missing data due to uncontrollable factors such as power supply failures."

Figure 4: To be consistent with the text, change "Angstrom" to "Ångström", and I don't know if the heights for Cases I and II should be reported to 1 decimal place, unless that is instrument precision (appears so e.g., from P14 Line 4)? Finally, I don't know how impactful this figure is visually if the x-axes are different for Case I and II. I would suggest either keeping x-axes consistent or overlapping the traces for the two cases in one plot. Differences aren't obvious in the current display.

**Response:** We have changed "Angstrom" to "Ångström". The accuracy of the height for the instrument is one decimal place. The more important role of Fig. 5 is to determine if the cases meet our prior selection criteria (section 3.2 in original manuscript), as was done in other studies (e.g., Fernández et al., 2015; Granados-Muñoz et al., 2015; Lv et al., 2017; Bedoya-Velásquez et al., 2018). Further research was conducted when there was sufficient reason to believe that the aerosol backscattering coefficient increase with height was primarily due to the increase in RH. Whether the x-axes are consistent does not affect our judgment. In addition, we have updated Fig. 5.

[Figure]

Fig. 5. The vertical profiles of (a, g) water vapor mixing ratio ($W$), (b, h) potential temperature ($\theta$), (c, i) relative humidity (RH) calculated from radiosonde data, (d, j) backscattering coefficient at 532 nm ($\beta_{532}$), (e, k) the Ångström exponent [AE (532-1064nm)], (f, l) depolarization ratio retrieved from Raman lidar data for Case I (top panels) and Case II (bottom panels). Horizontal dashed lines show the upper and lower boundaries of the layer under analysis (1642.5–1905.0 m for Case I and 1680.0–2130.0 m for Case II). Horizontal error bars denote the uncertainty of each property.

Figure 5: No major comments.

Figure 6: No major comments.

Figure 7: No major comments.

Figure 8: No major comments.

Figure 9: Please fix "Aerosol particle", but more importantly, in the caption, explain this is not data, but a model based on Eq. 3 (as per the text, unless I'm mistaken).

Done.

"Fig. 9. Aerosol size hygroscopic growth factor (GF) as a function of relative humidity (RH) for (a) Case I and (b) Case II. The different colors represent different particle sizes (Dp). These are the results of a model based on Eq. 3 from Gysel et al. (2009)."

Tables 1-3: No comments.

Table 5: The results weakly support the conclusions of the document. I would encourage the authors to be more transparent with their data, perhaps in a Supporting Information section. Any simple errors to report, e.g. 95% confidence intervals? How was the raw data from the H-TDMA obtained? A timeseries to serve as example perhaps?

**Re:** We decided to remove Table 5.

**Newly Added References:**

[revised manuscript text omitted]

---

## Author Comment (AC2) · 14 Nov 2018

**RESPONSES TO REVIEWER#2 COMMENTS**

**General Response:**

We appreciate the reviewer's comments on the manuscript entitled "Aerosol hygroscopic growth, contributing factors and impact on haze events in a severely polluted region in northern China". All comments are highly valuable and helpful for us to improve our manuscript. We have studied them carefully and have addressed them in the revised manuscript which includes additional investigations. Below are point-by-point responses to the reviewer's comments.

1. Add more descriptions or details on the instruments, e.g, what are measured directly and what are derived, uncertainties in their measured/derived quantities.

**Response:** The Raman lidar system directly measures atmospheric Mie scattering signals at 355, 532, and 1064 nm, and vibrational Raman scattering signals from $H_2O$ and $N_2$ molecules at 386 and 407 nm. Aerosol optical properties (aerosol extinction and backscattering coefficient, Ångström exponent, and the depolarization ratio) and atmospheric water vapor mixing ratio profiles can then be derived from this information. The errors of all parameters used in this study are now given in the revised manuscript.

[revised manuscript text omitted]

2. The four variables representing aerosol hygroscopicity from different aspects, namely the aerosol backscattering enhancement factor [$f$(RH)], the aerosol particle growth factor, the aerosol acidity and the hygroscopicity parameter. To what extent are these variables correlated? What are the correlations among $f$(RH) , and Äÿ and acidity? Under what circumstances? The answers would make the manuscript potentially more useful, e.g. for aerosol modeling. If possible, expand studied RH range, as I understand there is available data (below the selected loft layers and with lower RH) from the measurements.

**Response:** The aerosol size growth factor (GF) measures the change in particle diameter due to water uptake. Different from GF, $f$(RH) represents the aerosol backscattering coefficient hygroscopic enhancement factor of the aerosol population

and is mainly determined by the particle number size distribution (PNSD), chemical composition, and refractive index (Chen et al., 2014). The GF can affect $f$(RH) by changing the aerosol particle size and refractive index. On the one hand, the scattering cross-section of the aerosols will be enhanced due to aerosol size hygroscopic growth. On the other hand, aerosol size hygroscopic growth will reduce the refractive index, so the aerosol scattering efficiency becomes smaller. The relationship between the hygroscopicity parameter ($\kappa$) and GF can be expressed as follows:

$$\kappa = \frac{(GF^3 - 1)(1 - a_w)}{a_w} \quad , \tag{1}$$

where $a_w$ is the water activity. The Köhler equation, RH $= a_w \cdot S_k$, describes the equilibrium RH for a solution droplet, where $S_k$ is the Kelvin factor. When the particle size is larger than 100 nm, the Kelvin effect can be ignored, and $a_w$ in Eq. (1) can be replaced by RH. Chen et al. (2014) introduced a retrieval method to calculate $\kappa$ based on in situ measured $f$(RH) and PNSD obtained from the North China Plain. They showed that the $f$(RH) curves elevate as the mean $\kappa$ value increases. The aerosol acidity is a parameter that can affect aerosol hygroscopic growth, toxicity, and heterogeneity (Iinuma et al., 2004; Yao et al., 2007; Sun et al., 2010). When atmospheric aerosols are acidic, they are more strongly hygroscopic than in their neutralized form (Zhang et al., 2015). To the best of our knowledge, research on the simultaneous correlations among these four variables has not been carried out until now. We will do more detailed correlation studies in the future.

In reference to previous research (e.g., Fernández et al., 2015; Granados-Muñoz et al., 2015; Lv et al., 2017; Bedoya-Velásquez et al., 2018), a simultaneous increase in aerosol backscattering coefficient and RH values is the precondition for determining the range of heights and RHs considered in this study. Data below the selected layers aloft do not fulfill this requirement, so we cannot expand the RH range in this study. We will consider your comment in future work.

3. Figure 3 is an important figure for this manuscript, however I find it is hard to read or draw conclusions with it. And the description of Fig 3. is lack of clarity. Why and how the two cases are selected are poorly demonstrated in the text. Consider adding time series of surface water vapor mixing ratio and RH, as PM and chemical composition data are both obtained at the surface, and the authors are trying to draw some relationships between surface RH and these aerosol data.

**Response:** We have updated Figure 3. The associated discussion of the figure was also revised (see below).

[Figure]

Fig. 3. Time series of (a) water vapor mixing ratio ($W$) profiles measured by the Raman lidar, (b) mass concentrations of $PM_1$ (red dots) and $PM_{2.5}$ (blue dots), and (c) surface $W$ (black line) and relative humidity (RH, red line), and (d) chemical species mass fractions of $PM_1$ measured by the ACSM. Data are from 19–31 May 2016 at Xingtai. The shade grey areas are to enhance the readability of the article. The black triangles in (d) and grey lines in (a, b, c, d) represent the two cases chosen for further examination. Blank parts of the data are missing due to uncontrollable factors such as power supply.

**More description of why and how the two cases are selected:**

"As $W$ in the lower atmospheric layer and the surface mass concentrations of $PM_1$ and $PM_{2.5}$ increases, the proportion of organic aerosols decreases (highlighted as shaded grey areas in Fig. 3), suggesting that the proportion of hygroscopic aerosols increased. This shows that strong aerosol hygroscopicity may aggravate air pollution conditions over Xingtai. ""Two instances when this relationship was not seen (highlighted as shaded grey areas in Fig. 3) are shown by the black triangles in Fig. 3d and marked with grey lines across Fig. 3. In the evening of 21 May 2016 (the leftmost triangle and grey line), $W$ and the mass fractions of organics are comparable to those in the evening of

23 May (the rightmost triangle and grey line in Fig. 3). However, the mass concentrations of $PM_1$ and $PM_{2.5}$ at that time indicated by the leftmost grey line (in the evening of 21 May 2016) are significantly less than that in the evening of 23 May (indicated by the rightmost grey line). The cases occurring on 21 May 2016 (Case I) and 23 May 2016 (Case II) were selected for further study."

4. Page 9. Line 4-8, The authors give absolute errors of Raman-lidar-derived relative humidity and water vapor mixing ratio for a relatively dry case (20%<RH<35% ) in Fig 1 and 2. However boundary layers are generally wetter, and the two case selected for the study both occurred under atmospheric environments with RH>80%. What is the error of Raman-lidar derived RH for wet environment? It would be more meaningful to add a relatively wet case for validation. Also what are other uncertainties from Raman Lidar, e.g, AE, depolarization ratio? Is there any difference in uncertainty lower and higher altitudes?

**Response:** We have added a figure like Fig. 1 but representing a wet environment (the new Fig. 2; see below). Error bars are shown in Fig. 5. It is difficult to determine the uncertainties of the atmospheric aerosol backscattering coefficient, the AE, and the depolarization ratio retrieved by the lidar. This is because they are related to the height, the performance of the lidar system itself, weather conditions, and the properties of aerosols at that time, etc. The height and system performance mainly affect the signal-to-noise ratio of the returned signal. So the uncertainties of these parameters might vary with height.

[Figure]

Fig. 2. (a, c) Water vapor mixing ratio ($W$) and relative humidity (RH) profiles at 2000 BJT 23 May 2016 retrieved by the Raman lidar (blue line) and the radiosonde (red dashed line), respectively, and (b, d) the absolute error in $W$ and RH between the lidar and radiosonde retrievals (lidar minus radiosonde), respectively.

Other comments:

Page 7. Line 10-11, Please specify if the handheld particle/mass meter (PC-3016A) measures dry mass or total mass (including water uptake).

**Response:** The PC-3016A measures the total mass concentrations of $PM_1$ and $PM_{2.5}$, including water uptake, not just the dry mass.

Page 11. Line 12-13, "When AV=1.25, 50% of the total sulfate icons in the atmosphere consists of …and 50% consist of…" This sounds definite. Isn't this just a possible combination of different chemical components?

**Response:** These numbers were calculated using aerosol acidity and the pairing rule (Gysel et al., 2007). When $AV$=1.25, the mole number of the needed $NH_4^+$ to completely neutralize sulfate, nitrate, and chloride is 1.25 times than that of the measured $NH_4^+$. If the mole number of sulfate is 0.625 mol, the $NH_4^+$ paired with sulfate is 1 mol. The final calculation is: "50% of the total sulfate ions in the atmosphere consist of $NH_4HSO_4$, and the other 50% consist of $(NH_4)_2SO_4$".

Page 11 line 20, There are two "because" in this sentence, making it awkward.

**Response:** Fixed.

"The ACSM mainly measures the mass concentrations of $SO_4^{2-}$, $NO_3^-$, $NH_4^+$, $Cl^-$, and organics. The chlorine ion was not considered because its concentration is extremely low."

Page 13, line 3-5 the whole sentence, starting with "To see if this is the true", reads awkward.

**Response:** It has been rephrased in the revised manuscript.

Page 13, line 5-7, "As W in the lower atmospheric layer and the mass concentrations of PM1 and PM2.5 increased, the proportion of organic aerosols decreased, suggesting that the proportion of hygroscopic aerosols increased." This relationship is not straightforward by looking at Fig 3. Please think of a way of pointing to the readers where to look, maybe by marking these cases. Also in the next paragraph, two cases are selected. Consider adding two vertical lines across Fig a-b-c) so that data can be better visualized.

**Response:** The figure and corresponding paragraphs have been revised. See the reply to the third comment for details.

Page 13, line 8-18, This paragraph is related to Figure 3 and is lack of clarity. "this relationship" in the first sentence needs to be explicitly defined. Line 9 "relatively higher", what does it compare to? The first sentence implies the two cases are similar because "this relationship was not seen" in the two instances. However reading along, there seems to be differences for the two cases. What are the similarities and what are the differences for the two cases are not clearly stated in this paragraph. Why are they selected as the studied cases?

**Response:** Related paragraphs have been revised (see the reply to the third comment).

Page 14, line 4. How and why are these altitude ranges are selected? I understand that

the authors choose RH=80% as the reference RH. Why don't use the whole well mixed boundary layer, which can give a wider range of RH? I would be curious to see the hygroscopic growths under a wider range of RH. Would the regression relationships between RH and the various hygroscopic growth factors still be valid? If not, how much deviation there would be? Just thinking from aerosol modeling point of view, the result of this paper is potentially applicable in model parameterizations of aerosol hygroscopic growth if a wider RH range can be studied.

**Response:** As mentioned above, a simultaneous increase in aerosol backscattering coefficient and RH values is the precondition for determining the range of altitudes and RHs considered in this study (section 3.2 in original manuscript). In reference to previous studies (e.g., Fernández et al., 2015; Granados-Muñoz et al., 2015; Lv et al., 2017; Bedoya-Velásquez et al., 2018), the boundary layer height was not taken into account in the case selection. Your comments are sound, but the experimental data used in this study are limited and preclude doing what you suggest. We will consider your comment in future work.

12. Figure 7. If I understand correctly, the reverse of the slopes of fitted lines would be the Acid Value ($AV$). So maybe consider switch x and y axes for this figure. Then the slopes would be the $AV$ with no need to calculate the reverses.

**Response:** We switched the x- and y-axes in this figure as suggested. However, we think that the original figure looks better.

Page 14, 19-20, "This suggests that aerosol particles were transported to Xingtai from the same source region". Are there local emissions? Can it be excluded?

Response: The description has been revised as follows:

"This suggests that the aerosols in each case were transported into their respective layers at low speeds from almost the same direction. In other words, there is no change in the aerosol type of both cases within the region of interest."

Page 18. Line 5 "a positive result". Please be explicit.

Response: This means that the results between the remotely sensed and in situ measurements are consistent.

Page 20, line 2. I think the authors meant relative humidity by "water vapor content". This may have appeared in other places in the draft. Please don't mix use.

Response: Revised.

**Newly Added References:**

Chen, J., Zhao, C. S., Ma, N. and Yan, P.: Aerosol hygroscopicity parameter derived from the light scattering enhancement factor measurements in the North China Plain, Atmos. Chem. Phys., 14(15), 8105–8118, doi:10.5194/acp-14-8105-2014, 2014.

Ng, N. L., Herndon, S. C., Trimborn, A., Canagaratna, M. R., Croteau, P. L., Onasch, T. B., Sueper, D., Worsnop, D. R., Zhang, Q., Sun, Y. L. and Jayne, J. T.: An Aerosol Chemical Speciation Monitor (ACSM) for routine monitoring of the composition and mass concentrations of ambient aerosol, Aerosol Sci. Technol., doi:10.1080/02786826.2011.560211, 2011.

Stolzenburg M.R. and McMurry P. H.: Equations governing single and tandem DMA configurations and a new lognormal approximation to the transfer function, Aerosol Sci Tech, 42, 421–432, 2008.

Tan H., Xu H., Wan Q., Li F., Deng X., Chan P. W., Xia D. and Yin Y.: Design and application of an unattended multifunctional H-TDMA system, J. Atmos. Ocean Tech., 30, 1136–1148, doi:10.1175/JTECH-D-12-00129.1, 2013.